# Gradient Inversion Attack on Graph Neural Networks

**Divya Anand Sinha**     *dasinha@uci.edu*
*University of California, Irvine*

**Ruijie Du**     *ruijied@uci.edu*
*University of California, Irvine*

**Yezi Liu**     *yezil3@uci.edu*
*University of California, Irvine*

**Athina Markopolou**     *athina@uci.edu*
*University of California, Irvine*

**Yanning Shen**[*]     *yannings@uci.edu*
*University of California, Irvine*

**Reviewed on OpenReview:** *https://openreview.net/forum?id=aOmLrqkWyx*

## Abstract

Graph federated learning is of essential importance for training over large graph datasets while protecting data privacy, where each client stores a subset of local graph data, while the server collects the local gradients and broadcasts only the aggregated gradients. Recent studies reveal that a malicious attacker can steal private image data from the gradient exchange of neural networks during federated learning. However, the vulnerability of graph data and graph neural networks under such attacks, i.e., reconstructing both node features and graph structure from gradients, remains largely underexplored. To answer this question, this paper studies the problem of whether private data can be reconstructed from leaked gradients in both node classification and graph classification tasks and proposes a novel attack named Graph Leakage from Gradients (GLG). Two widely used GNN frameworks are analyzed, namely GCN and GraphSAGE. The effects of different model settings on reconstruction are extensively discussed. Theoretical analysis and empirical validation demonstrate that, by leveraging the unique properties of graph data and GNNs, GLG achieves more accurate reconstruction of both nodal features and graph structure from gradients.

## 1 Introduction

Federated Learning (McMahan et al., 2016b) is a distributed learning paradigm that has gained increasing attention. Gradient averaging is a widely used mechanism in federated learning, where clients send the gradients of the models instead of the actual private data to a central server. The server then updates the model by taking the average gradient over all the clients. The computation is executed independently on each client and synchronized via exchanging gradients between the server (Li et al., 2014; Iandola et al., 2016) and the clients (Patarasuk & Yuan, 2009). Various organizations and mobile devices in healthcare rely on federated learning to train models while also keeping user data private. For example, multiple hospitals train a model jointly without sharing their patients' medical records (Jochems et al., 2017; McMahan et al., 2016a).

Recently, several efforts have demonstrated that gradients, accessible to an honest-but-curious (HBC) server, can be exploited via gradient inversion attacks to reconstruct the original input data. By leveraging the gradients, Jeon et al. (2021); Zhu et al. (2019); Geiping et al. (2020); Zhao et al. (2020b); Yin et al. (2021);

---

[*]Corresponding Author.

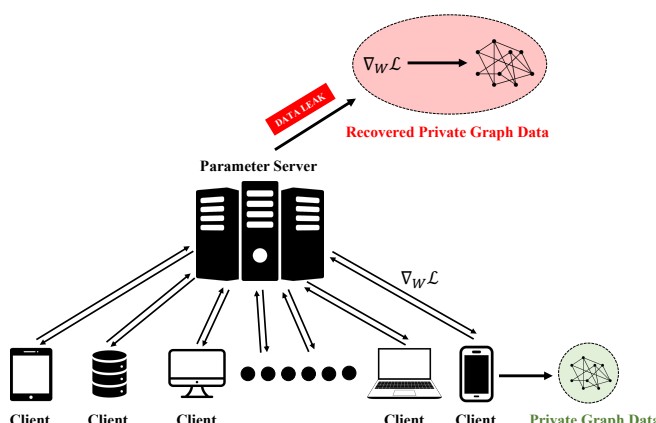

Figure 1: Gradient inversion attack in Federated Graph Learning

Zhu & Blaschko (2020); Jin et al. (2021) successfully reconstruct the users' private image data and Boenisch et al. (2021); Fowl et al. (2022a) reconstruct users' text data. Gradient inversion attacks have demonstrated the privacy vulnerabilities of standard federated learning.

Meanwhile, Graph Neural Networks (GNNs) have demonstrated state-of-the-art results when applied to structured data. From learning meaningful node representations over large social networks (Hamilton et al., 2017), to predicting molecular property (Gómez-Bombarelli et al., 2018; Yang et al., 2019), GNNs have been widely adopted across various paradigms. The confluence of Federated Learning and graphs has led to the emergence of Federated Graph Learning (FGL) (He et al., 2021; Scardapane et al., 2020; Zhang et al., 2021).

However, the privacy concerns regarding gradient leakage in FGL remain crucial and underexplored. Prior work has explored gradient-based attacks and privacy techniques in federated graph learning (Geisler et al., 2021; Ge et al., 2024), which aims to mislead training. Meanwhile, gradient inversion attacks have shown notable success in general deep neural networks (Zhu et al., 2019; Zhao et al., 2020a). Nevertheless, graph-specific gradient-inversion attacks, i.e., reconstructing both node features and graph structure from gradients, remain largely underexplored. Hence, the present work aims to answer the following underexplored research question:

> How can the unique properties of graph data and GNNs be leveraged to effectively reconstruct graph structure and node features from gradients of GNNs?

A major challenge in reconstructing graph data is the intertwined nature of the nodal features and the irregular graph structures, making it difficult to separate the information from nodal features and the graph structure. In other words, representations are correlated, and the underlying graph structure influences the correlation. In this work, we provide a theoretical analysis of the feasibility of reconstructing graph data based on gradients. Motivated by this observation, we propose a gradient inversion attack called GLG (Graph Leakage From Gradients). We investigate two commonly used GNN frameworks, i.e., Graph Convolutional Networks (GCN) (Kipf & Welling, 2016), and GraphSAGE (Hamilton et al., 2017), and examine their vulnerabilities to gradient inversion attacks for both node and graph classification tasks. An illustration of the attack scenario is presented in Figure 1, showcasing the process in which a curious central parameter server attempts to reconstruct the graph data utilizing the gradients of the model obtained from a client. **To our best knowledge, this is among the first works to study gradient-inversion attacks for graph data.** Our main contributions are summarized as follows:

- Theoretical analysis revealing the vulnerabilities of GNNs towards gradient inversion attack.
- Developing novel gradient inversion attack mechanisms to reconstruct graph data for both node and graph classification tasks.
- Experiments are conducted in a variety of settings with different amounts of prior knowledge.

The structure of the rest of the paper is as follows. Section 2 reviews related work. Section 3 introduces the problem formulation and the federated graph learning framework. Section 4 presents the proposed gradient inversion attack algorithms, GLG, and Section 5 provides the corresponding theoretical analysis. Section 6 presents the experimental results. Section 7 concludes the paper and discusses future work.

## 2 Related Work

**Gradient inversion attacks in FL.** In terms of privacy leakage, communicating gradients throughout the training process in FL can reveal sensitive information about the participants. Deep leakage from gradients (DLG)(Zhu et al., 2019) demonstrated that sharing the gradients can leak private training data, including images and text. Specifically, Zhao et al. (2020a) proposes that the ground truth labels can be directly inferred from the gradients. Consequently, ensuing works improve the attack techniques on large batches of user data (Yin et al., 2021; Fowl et al., 2022b). A few recent works study reconstructing text data in federated learning. To facilitate the text reconstruction, Boenisch et al. (2021); Fowl et al. (2022a) reconstruct text data using a strong threat model in which the server is malicious and is able to manipulate the training model's weights. In contrast, Gupta et al. (2022) studies the feasibility of reconstructing text from large batch sizes by leveraging the memorization capability of Language Models during federated learning. Recent work Petrov et al. (2024) introduces DAGER, the first exact gradient inversion attack for transformers by leveraging the low-rankness of self-attention layer gradients. The majority of the works are based on either image or text data.

**Gradient inversion attacks in FGL.** Reconstructing the data, including nodal features and graph structure, in FGL remains underexplored compared to the gradient inversion attacks in image or text domains. Since the input of GNN includes nodal features and graph structure, directly applying DLG or its enhanced methods in the graph domain is either infeasible or leads to significant performance degradation. Particularly in GCN, DLG consistently struggles to reconstruct the graph structure, which is essential for graph data. The only concurrent work, Drencheva et al. (2025), proposes GRAIN that introduces a gradient inversion attack tailored to GCN (Kipf & Welling, 2016) and GAT (Veličković et al., 2018). GRAIN employs a depth-first search strategy to identify subgraphs within gradients, iteratively filtering out unlikely candidates through a span check. By leveraging degree information in the features, GRAIN generates its final prediction. However, nodal degree information is not inherently accessible to attackers during GNN training. In the absence of degree information, neither theoretical nor experimental guarantees of GRAIN can keep reliable. Given the limited exploration of gradient inversion attacks in FGL, we propose a method, GLG, that reconstructs both nodal features and graph structure with fewer constraints. GLG does not rely on explicit degree information, making it more generalizable. To facilitate comparison, we summarize the key differences between DLG, GRAIN, and our method in Table 1. The detailed comparison is provided in Appendix H.

Table 1: Comparison of DLG, GRAIN and GLG. ✓and ✗indicate whether the method can reconstruct the corresponding graph data. Prior knowledge refers to any other information required besides the gradients.

| Method | Nodal feature | Graph structure | Prior knowledge |
|--------|:---:|:---:|:---:|
| DLG | ✓ | ✗ | None |
| GRAIN | ✓ | ✓ | Node degree |
| GLG | ✓ | ✓ | None |

## 3 Preliminaries

Given a graph $\mathcal{G} := (\mathcal{V}, \mathcal{E})$, where $\mathcal{V} := \{v_1, v_2, \cdots, v_N\}$ represents the node set and $\mathcal{E} \subseteq N \times N$ denotes the edge set. Matrices $\mathbf{X} \in \mathbb{R}^{N \times D}$ and $\mathbf{A} \in \{0, 1\}^{N \times N}$ represent the node feature and adjacency matrix respectively, where $\mathbf{A}_{ij} = 1$ if and only if $\{v_i, v_j\} \in \mathcal{E}$. $\tilde{\mathbf{A}} \in \mathbb{R}^{N \times N}$ denotes the normalized adjacency matrix derived from $\mathbf{A}$. In a GCN framework $\tilde{\mathbf{A}} = \mathbf{D}^{-\frac{1}{2}}(\mathbf{A} + \mathbf{I})\mathbf{D}^{-\frac{1}{2}}$ where $\mathbf{D} \in \mathbb{R}^{N \times N}$ is the diagonal degree matrix with $i$-th diagonal entry $d_i$ denoting the degree of node $v_i$. While for GraphSAGE with mean aggregation, $\tilde{\mathbf{A}} = \mathbf{D}^{-1}\mathbf{A}$. Let $\mathbf{L} = \mathbf{I} - \mathbf{D}^{-\frac{1}{2}}\mathbf{A}\mathbf{D}^{-\frac{1}{2}}$ denote the symmetric normalized laplacian matrix. In a

node classification task, the node to be classified is called the target node, denoted as $v_n$, with $\mathbf{x}_{v_n}$ and $\mathcal{N}_{v_n}$ representing its feature vector and neighboring index set, respectively. Node representations at the $l$-th layer are denoted by $\mathbf{H}^l \in \mathbb{R}^{N \times F}$ where the $i$-th row of $\mathbf{H}^l$ i.e. $\mathbf{h}^l_{v_i} \in \mathbb{R}^{1 \times F}$ denotes the representation for the node $v_i$. Throughout the paper we use the operator $(\cdot)_i$ to denote the $i$-th row of any matrix or the $i$-th element of any vector, for example, $(\mathbf{X})_i$ denotes the $i$-th row of the node feature matrix $\mathbf{X}$.

### 3.1 GNN Frameworks

We first introduce GraphSAGE and Graph Convolutional Network (GCN) frameworks for both node and graph classification tasks below.

#### 3.1.1 Node Classification:

Consider a GraphSAGE layer with weights $\mathbf{W}_1 \in \mathbb{R}^{F \times D}, \mathbf{W}_2 \in \mathbb{R}^{F \times D}$, and bias $\mathbf{b} \in \mathbb{R}^{1 \times F}$ Let $\mathbf{h}^l_{v_n}$ and $\mathbf{h}^l_{\text{agg}}$ denote the hidden and aggregated representation of the node $v_n$ at layer $l$ and $\sigma$ is a non-linear activation function. Then the GraphSAGE layer can be expressed as

$$\mathbf{h}^l_{v_n} = \sigma(\mathbf{h}^{l-1}_{\text{agg}} \mathbf{W}^\top_1 + \mathbf{h}^{l-1}_{v_n} \mathbf{W}^\top_2 + \mathbf{b}), \tag{1}$$

with $\mathbf{h}^0_{v_n} = \mathbf{x}_{v_n}$ and $\mathbf{h}^0_{\text{agg}} = \mathbf{x}^{\text{agg}}_{v_n}$ denoting the feature vector and aggregated representation of node $v_n$ respectively. For a GraphSAGE layer with mean aggregation the aggregated representation of a node $\mathbf{h}^{l-1}_{\text{agg}}$ can be obtained as

$$\mathbf{h}^{l-1}_{\text{agg}} = \text{mean}_{j \in \mathcal{N}_{v_n}}(\mathbf{h}^{l-1}_{v_j}). \tag{2}$$

Similarly, for a GCN layer with weights $\mathbf{W} \in \mathbb{R}^{F \times D}$ and bias $\mathbf{b} \in \mathbb{R}^{1 \times F}$, the representation of a target node $v_n$ can be obtained via

$$\mathbf{h}^l_{v_n} = \sigma(\mathbf{h}^{l-1}_{\text{agg}} \mathbf{W}^\top + \mathbf{b}). \tag{3}$$

Here $\mathbf{h}^{l-1}_{\text{agg}}$ is obtained by aggregating the neighboring nodal feature vectors as

$$\mathbf{h}^{l-1}_{\text{agg}} = \sum_{j \in \mathcal{N}_{v_n} \cup \{n\}} \frac{\mathbf{h}^{l-1}_{v_j}}{\sqrt{d_n d_j}}. \tag{4}$$

For a node classification task using an $L$ layer GraphSAGE or GCN framework let $K$ denote the number of classes. Also, let $\mathbf{h}^L_{v_n} \in \mathbb{R}^{1 \times K}$ denote the output of the final layer and $\mathbf{p} = \mathbf{h}^L_{v_n}$. The final Cross-Entropy loss of a node $v_n$ can be written as

$$\mathcal{L}_{v_n} = -\log \frac{e^{\mathbf{p}_k}}{\sum_{j=1}^K e^{\mathbf{p}_j}}, \tag{5}$$

where $k$ is the corresponding ground-truth label and $\mathbf{p}_j$ denotes the $j$th element of the vector $\mathbf{p}$.

#### 3.1.2 Graph Classification:

Consider an $N$-node graph with $\mathbf{X}$ and $\tilde{\mathbf{A}}$ as the node feature matrix and the normalized adjacency matrix respectively, let $\mathbf{H}^l \in \mathbb{R}^{N \times F}$ denote the hidden representations of the nodes at layer $l$. The output of a GraphSAGE layer can be written as

$$\mathbf{H}^l = \sigma(\tilde{\mathbf{A}}\mathbf{H}^{l-1}\mathbf{W}^\top_1 + \mathbf{H}^{l-1}\mathbf{W}^\top_2 + \vec{\mathbf{1}}\mathbf{b}), \tag{6}$$

with $\mathbf{H}^0 = \mathbf{X}$. Here $\vec{\mathbf{1}}$ denotes a column vector of all ones. Similarly, the output of a GCN layer can be written as

$$\mathbf{H}^l = \sigma(\tilde{\mathbf{A}}\mathbf{H}^{l-1}\mathbf{W}^\top + \vec{\mathbf{1}}\mathbf{b}). \tag{7}$$

Following the final GNN layer, an MLP layer is applied . For an MLP with weights $\mathbf{W} \in \mathbb{R}^{K \times NF}$, the final readout from the MLP layer can be written as

$$\mathbf{h}_{\text{MLP}} = \mathbf{h}_f \mathbf{W}^\top + \mathbf{b}. \tag{8}$$

where $\mathbf{h}_f \in \mathbb{R}^{1 \times NF}$ is the flattened representation of $\mathbf{H}^L \in \mathbb{R}^{N \times F}$ which is the hidden representations obtained at the $L$th layer. Let $\mathbf{p} = \mathbf{h}_{\text{MLP}} \in \mathbb{R}^{1 \times K}$ be the final readout, the Cross-Entropy Loss $\mathcal{L}$ can be computed using equation 5.

## 3.2 Federated Graph Learning (FGL)

A typical FL framework consists of a server and $C$ clients. In the node-level FGL, each client samples a mini-batch of $B$ target nodes $\{\mathbf{x}_{v_1}, \cdots, \mathbf{x}_{v_B}\}$ from the graph which is used to compute the gradients of the loss function in equation 5 with respect to the model parameters. In the graph-level FGL, at each step $t$, each client has a set of multiple graphs. Similar to the node-level FGL the $c^{th}$ client samples a minibatch of $B$ graphs $\{\mathcal{G}_{c_1}, \cdots, \mathcal{G}_{c_B}\}$ which are then used to compute the gradients and train the model. The gradients from $N$ clients are then broadcast to the server for updating the weights as

$$\nabla_{\mathbf{W}^t} \mathcal{L} = \frac{1}{B \cdot C} \sum_{c=1}^{C} \sum_{i=1}^{B} \nabla_{\mathbf{W}^t} \mathcal{L}_{c,i}, \tag{9}$$

$$\mathbf{W}^{t+1} = \mathbf{W}^t - \eta \nabla_{\mathbf{W}^t} \mathcal{L} \tag{10}$$

where $\nabla_{\mathbf{W}^t} \mathcal{L}_{c,i}$ denotes the gradient of the loss at the $c$-th client for the $i$-th data sample.

## 4 Gradient Inversion Attack for GNNs

We consider the scenario where there is an honest-but-curious server, i.e., the server has access to the model gradient with respect to the client's private graph data. Different attack scenarios are next discussed, where the server may also have access to different amounts of information.

### 4.1 Threat Model

The attacker is an *honest-but-curious* server that aims to reconstruct the user's graph data given the gradients of a GNN model. In addition to possessing knowledge of the gradients, the attacker may also have access to varying amounts of information about the user's private graph data. Hence we categorize the attacks based on the task as follows:

**Node Attacker 1**: has access to the gradients of a node classification model for the target node and reconstructs the nodal features of the target node and its neighbors.

**Node Attacker 2**: has access to the gradients of a node classification model for all the nodes in an egonet or subgraph. Specifically, the gradients are obtained for the loss function $\mathcal{L}_v \in \mathbb{R}^{1 \times N}$ with respect to the neural networks weights, where each element $\mathcal{L}_{v_i}$ denotes the loss for node $v_i$. Unlike traditional classification tasks, the adjacency information is also significant for graph data. In this setting, the graph structure could be unknown. The honest-but-curious server would also try to reconstruct the nodal features and/or the graph structure. Based on the input, we categorize the attacker furhter as follows

- **Node Attacker 2-G**: has access to the gradients of all the nodes in an egonet or subgraph along with the nodal features and reconstructs the underlying graph structure.
- **Node Attacker 2-N**: has access to the gradients of all the nodes in an egonet or subgraph along with the graph structure and reconstructs the nodal features.
- **Node Attacker 2-GN**: has access only to the gradients of all nodes in an egonet or subgraph and reconstructs the graph structure as well as the nodal features.

**Graph Attacker**: has access to the gradients of the graph for a graph classification model. Similar to Node Attacker 2, in this setting the honest-but-curious server would try to reconstruct the nodal features and/or the graph structure. Hence, we categorize the attack in more detail as follows

- **Graph Attacker-G**: has access to the gradients along with the nodal features and reconstructs the underlying graph structure.
- **Graph Attacker-N**: has access to the gradients along with the graph structure and reconstructs the nodal features.
- **Graph Attacker-GN**: has access to the gradients and reconstructs both the nodal features and the graph structure.

Table 2: Overview of Threat Models: distinct attackers across seven settings. $(\mathbf{X})_{target}$ refers to nodal features of the target node and its neighbors. $\nabla_{\mathbf{W}}$ denotes the gradients of nodes, $\mathbf{X}$ represents features of all nodes in the subgraph, and $\mathbf{A}$ corresponds to the underlying graph structure.

| Attacker | setting | Training task | Accessible information | Reconstructed data |
|---|---|---|---|---|
| Node Attacker 1 | | Node classification | $\nabla_{\mathbf{W}}$ of the target node | $(\mathbf{X})_{target}$ |
| Node Attacker 2 | G | Node classification | $\nabla_{\mathbf{W}}$ and $\mathbf{X}$ | $\mathbf{A}$ |
| Node Attacker 2 | N | Node classification | $\nabla_{\mathbf{W}}$ and $\mathbf{A}$ | $\mathbf{X}$ |
| Node Attacker 2 | GN | Node classification | $\nabla_{\mathbf{W}}$ | $\mathbf{X}$ and $\mathbf{A}$ |
| Graph Attacker | G | Graph classification | $\nabla_{\mathbf{W}}$ and $\mathbf{X}$ | $\mathbf{A}$ |
| Graph Attacker | N | Graph classification | $\nabla_{\mathbf{W}}$ and $\mathbf{A}$ | $\mathbf{X}$ |
| Graph Attacker | GN | Graph classification | $\nabla_{\mathbf{W}}$ | $\mathbf{X}$ and $\mathbf{A}$ |

## 4.2 Attack Mechanisms

In this section, GLG is introduced to attack a GNN framework and steal private user data using the gradients in a federated graph learning setting. Specifically, given the gradients of the model from a client for the $i$-th data sample $\nabla_{\mathbf{W}}\mathcal{L}_{c,i}$, the goal is to reconstruct the input graph data sample that was fed to the model. For simplicity, we denote $\nabla_{\mathbf{W}}\mathcal{L}_{c,i}$ as $\nabla_{\mathbf{W}}\mathcal{L}$ for the graph classification task, and as $\nabla_{\mathbf{W}}\mathcal{L}_v$ for the node classification task. In Zhu et al. (2019), the input data to a model is reconstructed by matching the dummy gradients with the gradients. The dummy gradients are obtained by feeding dummy input data to the model. Given the gradients at a certain time step as $\nabla_{\mathbf{W}}\mathcal{L}$, the input data is reconstructed by minimizing the following objective function

$$\mathbb{D} = ||\nabla_{\mathbf{W}}\hat{\mathcal{L}} - \nabla_{\mathbf{W}}\mathcal{L}||^2. \tag{11}$$

Here $\nabla_{\mathbf{W}}\hat{\mathcal{L}}$ is the gradient of the loss obtained from the dummy input data. In Geiping et al. (2020) the cosine loss optimization function is used instead of the $\ell_2$ loss in equation 11. Specifically, the objective is to minimize the cosine loss between the actual gradients shared by a client and the gradients obtained from dummy input data

$$\mathbb{D} = 1 - \frac{\nabla_{\mathbf{W}}\hat{\mathcal{L}} \cdot \nabla_{\mathbf{W}}\mathcal{L}}{||\nabla_{\mathbf{W}}\hat{\mathcal{L}}||||\nabla_{\mathbf{W}}\mathcal{L}||}. \tag{12}$$

In this work, it is assumed that the input label is known since in a classification task with cross-entropy loss, the labels can be readily inferred from the gradients (Zhao et al., 2020a). For further details regarding this please refer to Appendix B.

**Feature Smoothness.** As shown in equations 2 and 4 Graph Neural Networks (GNNs) typically employ a feature aggregation mechanism wherein node representations are constructed by integrating features from neighboring nodes prior to weight multiplication at each network layer. However, the objective function proposed in equation 12 may be insufficient for simultaneously reconstructing both nodal features and graph structure. To address this limitation, we leverage intrinsic properties characteristic of real-world graphs that can enhance graph data reconstruction. Many real-world graph structures, such as social networks, exhibit a fundamental property of feature smoothness, wherein connected nodes tend to possess similar characteristics. To formally enforce and quantify this feature smoothness in the reconstructed graph data, we propose the

following loss function, which is adapted from the approach introduced by Zhang et al. (2022):

$$\mathcal{L}s = tr(\mathbf{X}\mathbf{L}\mathbf{X}^\top)$$
$$= \sum i,j \in \mathcal{E}\left(\frac{\mathbf{x}_{v_i}}{\sqrt{d_i}} - \frac{\mathbf{x}_{v_j}}{\sqrt{d_j}}\right) \tag{13}$$

The loss function captures the similarity between adjacent nodes by minimizing the normalized feature differences across connected nodes.

In addition, the Frobenius norm regularizer is also introduced such that the norm of $\mathbf{A}$ is bounded. Overall, the final objective can be written as

$$\hat{\mathbb{D}} = \mathbb{D} + \alpha\mathcal{L}_s + \beta||\mathbf{A}||_F^2 \tag{14}$$

where $\mathbb{D}$ is defined in equation 12, $\alpha$ and $\beta$ are hyperparameters. The iterative algorithms for reconstructing the private data from the gradients in a node classification task and a graph classification task are summarized in algorithms 1, 2, and 3, in Appendix A, respectively. Specifically, Node Attacker 1 utilizes algorithm 1 for reconstructing the nodal features of the target node and its neighbors. Node Attackers 2 (G, N and GN) utilizes algorithm 2 for reconstructing the nodal features and graph structure given the gradients of the loss function for each node in the egonet/subgraph. By default, algorithm 2 assumes that both nodal features and graph structure are unknown and tries to reconstruct both. However, only the unknown parameter will be optimized when either of the two is known. For instance, Node Attacker 2-G only optimizes for $\mathbf{A}$ in Line 9 of algorithm 2 while skipping Line 8 since the nodal features $\mathbf{X}$ is known. The same holds for Node Attacker 2-N. However, unlike the other two Node Attacker 2-GN optimizes for both. Similar logic holds for Graph Attacker (a, b and c) which utilizes algorithm 3. Note that algorithms 2 and 3 might seem similar since they are reconstructing the same variables. However, a key difference between the two is that algorithm 2 is attacking a node classification model while algorithm 3 attacks a graph classification model. In algorithm 2, the gradients of the loss function are computed individually for each node. This is because each node within the subgraph has an associated label. In contrast, algorithm 3 calculates gradients for the entire graph, as the graph has a single label.

**Reconstructing the Adjacency Matrix:** In all the scenarios where the attacker is trying to reconstruct the adjacency matrix, projected gradient descent is applied. Specifically, the gradient descent step in Line 9 of algorithms 2 and 3 entails a projection step, where each entry $(i, j)$ of the adjacency matrix is updated through the entry-wise projection operator defined as

$$\hat{\mathbf{A}}_{ij} = \mathrm{proj}_{[0,1]}(\tilde{\mathbf{A}}_{ij}) = \begin{cases} 1, & \tilde{\mathbf{A}}_{ij} \geq 1 \\ 0, & \tilde{\mathbf{A}}_{ij} \leq 0 \\ \tilde{\mathbf{A}}_{ij}, & \text{otherwise} \end{cases} \tag{15}$$

Finally, to reconstruct the binary adjacency matrix, we consider each entry $\hat{\mathbf{A}}_{ij}$ as the probability of any edge between nodes $v_i$ and $v_j$ and the reconstructed binary adjacency matrix is obtained by sampling from a Bernoulli distribution with the corresponding probability at the last iteration, see also Line 11 of algorithm 2 and algorithm 3.

Since the reconstructed adjacency matrix $\hat{\mathbf{A}}$ is obtained via sampling through Bernoulli distribution, the Frobenius regularizer has the effect of reducing the magnitude of entries of $\hat{\mathbf{A}}$. Consequently, this regularizer also contributes to the promotion of sparsity in the ultimately reconstructed adjacency matrix. This sparsity property is particularly relevant since real-world social networks are sparse.

## 5 Theoretical Justification

While prior work (Geiping et al., 2020) has shown that the input to a fully-connected layer can be reconstructed from the layer's gradients, the analysis of GNNs is largely under-explored. Below we will extensively study whether similar conclusions apply to GraphSAGE and GCN and what parts of the graph input can be reconstructed from the gradients.

### 5.1 Node Attacker

Consider a node classification setting, where the inputs to a GNN layer are the target nodal features and the neighboring nodal features denoted as $\mathbf{x}_v$ and $\{\mathbf{x}_{v_j}\}_{j \in \mathcal{N}_v}$. Through theoretical analysis we will show that parts of the input to a GCN or GraphSAGE layer can be reconstructed from the gradients analytically without solving an iterative optimization problem.

**Proposition 1.** *For a GraphSAGE defined in equation 1 or a GCN layer defined in equation 3, Node Attacker 1 can reconstruct the aggregated representations of a target node denoted as $\mathbf{x}_v^{agg}$ (see equation 4 with $\mathbf{h}_{agg}^0 = \mathbf{x}_v^{agg}$ ) given the gradients of the first layer as $\mathbf{x}_v^{agg} = \nabla_{(\mathbf{W})_i}\mathcal{L}_v/\nabla_{(\mathbf{b})_i}\mathcal{L}_v$ for GCN and as $\mathbf{x}_v^{agg} = \nabla_{(\mathbf{W}_1)_i}\mathcal{L}_v/\nabla_{(\mathbf{b})_i}\mathcal{L}_v$ for GraphSAGE provided that $\nabla_{(\mathbf{b})_i}\mathcal{L}_v \neq 0$.*

*Proof.* We show the proof for a GCN layer. For a GCN layer, the aggregated input at the target node can be reconstructed from the gradients with respect to the weights of the first layer by writing the following equations

$$\mathbf{h}_v = \sigma(\tilde{\mathbf{h}}_v), \tag{16}$$

$$\tilde{\mathbf{h}}_v = \mathbf{x}_v^{\text{agg}}\mathbf{W}^\top + \mathbf{b}, \tag{17}$$

$$\nabla_{(\mathbf{W})_i}\mathcal{L}_v = \nabla_{(\tilde{\mathbf{h}}_v)_i}\mathcal{L}_v \cdot \nabla_{(\mathbf{W})_i}(\tilde{\mathbf{h}}_v)_i, \tag{18}$$

$$\nabla_{(\mathbf{W})_i}\mathcal{L}_v = \nabla_{(\mathbf{b})_i}\mathcal{L}_v \cdot \mathbf{x}_v^{\text{agg}} \tag{19}$$

where $(\mathbf{W})_i$ denotes the $i$th row of $\mathbf{W}$, $(\mathbf{b})_i$ and $(\tilde{\mathbf{h}}_v)_i$ denotes the $i$th element of vectors $\mathbf{b}$ and $\tilde{\mathbf{h}}_v$ respectively. It can be concluded from equation 30 that $\mathbf{x}_v^{\text{agg}}$ can be reconstructed as $\nabla_{(\mathbf{W})_i}\mathcal{L}_v/\nabla_{(\mathbf{b})_i}\mathcal{L}_v$ as long as $\nabla_{(\mathbf{b})_i}\mathcal{L}_v \neq 0$. □

It can be observed from the above proof that the analytic reconstruction is independent of the loss function or the non-linearity used after the layer and only depends on the gradients of the loss function with respect to the weights of the model.

**Proposition 2.** *For a GraphSAGE layer defined in equation 1, Node Attacker 1 can reconstruct the target node features given the gradients of the first layer as $\mathbf{x}_v = \nabla_{(\mathbf{W}_2)_i}\mathcal{L}_v/\nabla_{(\mathbf{b})_i}\mathcal{L}_v$ as long as $\nabla_{(\mathbf{b})_i}\mathcal{L}_v \neq 0$.*

*Proof.* See Appendix D. □

In the following, we focus on theoretically analyzing the attack setting of Node Attacker 2, specifically targeting the reconstruction of nodal features and/or the graph structure. To reiterate, this setting involves leveraging the gradients of the loss function of GNNs associated with each node in the egonet or subgraph.

**Proposition 3.** *For a GCN layer defined in equations equation 3 and equation 4, Node Attacker 2-G can reconstruct $\tilde{\mathbf{A}}$ given the gradients of the first layer as long as $\mathbf{X}$ is full row-rank.*

*Proof.* From Proposition 1 we know that given the gradients for a node we can reconstruct the aggregated representations of that node. Since we know the gradients for each node we can reconstruct the aggregated representation for each node. Let $\mathbf{X}^{\text{agg}} \in \mathbb{R}^{N \times D}$ denote the aggregated node representation matrix where each row $i$ denotes the aggregated node representation of node $v_i$, equation 4 in the matrix form can be written as the following for $l - 1 = 0$.

$$\mathbf{X}^{\text{agg}} = \tilde{\mathbf{A}}\mathbf{X}. \tag{20}$$

Since $\mathbf{X}^{\text{agg}}$ and $\mathbf{X}$ are known, the adjacency matrix can be then reconstructed as $\tilde{\mathbf{A}} = \mathbf{X}^{\text{agg}}\mathbf{X}^\dagger$. □

**Proposition 4.** *For a GCN layer defined in equations equation 4 and equation 3, Node Attacker 2-N can reconstruct $\mathbf{X}$ given the gradients of the first layer if $\tilde{\mathbf{A}}$ is full column-rank.*

*Proof.* Similar to Proposition 3 once we have $\mathbf{X}^{\text{agg}}$, $\mathbf{X}$ can be obtained as $\mathbf{X} = \mathbf{X}^{\text{agg}}\tilde{\mathbf{A}}^\dagger$ □

Table 3: Target node feature reconstruction $\mathbf{x}_v$ (RNMSE) for Node Attacker 1.

|  |  | Synthetic | FB | GitHub | OGBN-Arxiv |
|---|---|---|---|---|---|
| **GLG (Ours)** |  |  |  |  |  |
| GraphSAGE $(\times 10^{-3})$ | Mean | 2.8±2.5 | 2.6±1.3 | 2.8±1.4 | 4.4±3.5 |
|  | Min | 0.75 | 0.6 | 0.9 | 1.53 |
| GCN | Mean | 0.54±0.35 | 0.81±0.61 | 1.27±0.79 | 4.28±0.49 |
|  | Min | 0.23 | 0.22 | 0.18 | 3.25 |
| **DLG** |  |  |  |  |  |
| GraphSAGE | Mean | 0.113±0.07 | 0.03±0.06 | 0.001±0.0012 | 4.57±0.37 |
|  | Min | 0.04 | 0.001 | 0.0009 | 3.92 |
| GCN | Mean | 3.21±2.56 | 1.34±1.66 | 7.01±5.49 | 4.98±1.11 |
|  | Min | 1.22 | 0.14 | 1.51 | 3.64 |

**Proposition 5.** *For a GraphSAGE layer defined in equation 1, Node Attacker 2-GN can reconstruct both* $\mathbf{X}$ *and* $\tilde{\mathbf{A}}$ *given only the gradients of the first layer for each node.*

*Proof.* Proposition 2 states that for a GraphSAGE layer, the target node features can be reconstructed given the gradients of the first layer for that node. Similarly, if we have access to the gradients for each node in the graph we can reconstruct all the nodal features. Let $\mathbf{X}$ denote this matrix. From Propositions 1 it is known that for a GraphSAGE layer, the aggregated node representations $\mathbf{X}^{\mathrm{agg}}$ can be reconstructed. With both $\mathbf{X}$ and $\mathbf{X}^{\mathrm{agg}}$ in hand, the adjacency matrix can be reconstructed as $\tilde{\mathbf{A}} = \mathbf{X}^{\mathrm{agg}}\mathbf{X}^{\dagger}$ similar to Proposition 3. □

The aforementioned proposition has significant privacy implications. Even without any prior knowledge of the graph data, the attacker can successfully reconstruct both the nodal features $\mathbf{X}$ and the underlying graph structure $\mathbf{A}$. This result is particularly important as one would anticipate that the intertwined nature of graph data makes it impossible to independently reconstruct $\mathbf{X}$ and $\mathbf{A}$, which is however shown not to be true.

### 5.2 Graph Attacker

The present subsection provides theoretical analysis for a Graph Attacker-G in a scenario where prior information may be available for the graph data to be attacked. It can be shown that for GraphSAGE, prior knowledge of the nodal feature matrix $\mathbf{X}$ can help reconstruct the graph structure $\mathbf{A}$, making the framework more vulnerable to the gradient-inversion attacks.

**Proposition 6.** *For a GraphSAGE framework defined in equation 6 with* $\mathbf{H}^0 = \mathbf{X}$, *Graph Attacker-G can reconstruct* $\tilde{\mathbf{A}}$ *from the gradients of the first layer as* $\tilde{\mathbf{A}} = \mathbf{X}(\nabla_{\mathbf{W}_2}\mathcal{L})^{\dagger}(\nabla_{\mathbf{W}_1}\mathcal{L})\mathbf{X}^{\dagger}$ *if the nodal feature matrix* $\mathbf{X}$ *is full-row rank and known.*

*Proof.* See Appendix E. □

Therefore if the attacker knows the nodal feature matrix $\mathbf{X}$ along with the gradients, it can reconstruct the normalized adjacency matrix $\tilde{\mathbf{A}}$ analytically.
Even though it is not possible to analytically demonstrate the reconstruction for Graph Attacker-N and Graph Attacker-GN, we do conduct experiments for all the threat models for both GraphSAGE and GCN.

## 6 Data & Experiments

The effectiveness of the proposed algorithms is evaluated on real-world social network datasets, paper citation network datasets, molecular datasets, and synthetic datasets. The detailed results for molecular datasets are provided separately in Appendix G.

**Evaluation Metrics.** Error metrics for reconstructing the nodal features and the graph structure are evaluated, along with their standard deviation.

- **Nodal Features**: To evaluate the performance of the reconstructed node features, the Root Normalised Mean Squared Error(RNMSE) is used

$$\text{RNMSE}(\mathbf{x}_v, \hat{\mathbf{x}}_v) = \frac{||\mathbf{x}_v - \hat{\mathbf{x}}_v||}{||\mathbf{x}_v||}. \tag{21}$$

  In order to evaluate the performance of reconstructing $\mathbf{X}$ in a graph classification setting, we report the mean RNMSE over all the nodal features in $\mathbf{X}$.

- **Graph Structure**: To evaluate the reconstruction performance of the binary adjacency matrix, we use the following metrics

  - **Accuracy**: defined as $\frac{\sum_{i,j}(\hat{\mathbf{A}}_{ij} - \mathbf{A}_{ij})}{N*N}$.
  - **AUC** (Area Under Curve): The area under the Reciever Operator Characteristic Curve.
  - **AP** (Average Precision): is defined as $\frac{TP}{TP+FP}$ where $TP$ stands for number of True Positives and $FP$ stands for False Positives. True Positives are the correctly identified edges in the actual graph by the attacker and False Positives are the non-edges incorrectly classified as edges by the attacker.

**Experimental Settings.** Algorithms 1, 2 and 3 in Appendix A solve an optimization problem using gradient descent. The gradient descent step can be replaced with off-the-shelf optimizers such as Adam or L-BFGS. In all the experiments, Adam is used as the optimizer. In an attack setting, allowing multiple restarts to the optimizer (especially L-BFGS) from different starting points can significantly increase the quality of reconstruction. For a fair comparison, multiple restarts are not used, and all results presented are obtained with a single run of the optimizer. However, in an actual attack setting, an attacker may be able to *greatly improve* the reconstruction quality by allowing multiple restarts.

In the experiments, a 2-layer GNN model with hidden dimension 100 and a sigmoid activation function is employed. The weights of the model are randomly initialized. For all the experiments, the dummy nodal features are initialized randomly, with each entry sampled from the standard normal distribution i.e., $\mathcal{N}(0,1)$. The dummy adjacency matrix is initialized by randomly setting its entries to 0 or 1. The hyperparameters values for the feature smoothness and the sparsity regularizers are set to $\alpha = 10^{-9}$ and $\beta = 10^{-7}$. These are the values selected by grid search for the choice that leads to the best performance.

- **Node Attacker 1** uses the cosine loss objective given in equation 12 without regularization.
- **Node Attacker 2** uses the objective function is defined in equation 14.
- **Graph Attacker** uses the objective function as defined in equation 14.

We adopt the DLG attack as our baseline and evaluate it across all three settings. A comprehensive comparison with GRAIN is provided in Appendix H. We also extend the experiments involving Node Attacker 1 and Graph Attacker-N to the minibatch setting; these results are detailed in Appendix F.2. As the attack involves numerous hyperparameters, we further analyze a subset of these hyperparameters under various settings, with results presented in Appendix F.3. All reported results represent averages taken over 20 independent runs of each attack.

**Datasets.** We consider the following three datasets:

- **GitHub** (Rozemberczki et al., 2021): Nodes represent developers on GitHub and edges are mutual follower relationships. The nodal features are extracted based on the location, repositories starred, employer and e-mail address. Binary labels indicate the user's job title, either web developer or machine learning developer. The details regarding the number of nodes and edges in this dataset are presented in Table 10.

- **FacebookPagePage (FB)** (Rozemberczki et al., 2019): A page-page graph of verified Facebook sites. Nodes correspond to official Facebook pages, links represent mutual likes between sites. Node features are extracted from the site descriptions. The labels denote the categories of the sites. The details regarding the number of nodes and edges in this dataset are presented in Table 10.
- **OGBN-Arxiv** (Hu et al., 2020): A large-scale citation network dataset from the Open Graph Benchmark (OGB), representing academic papers from the arXiv repository. Each node corresponds to a paper, and directed edges represent citation links between them. Node features are based on the paper's title and abstract. The task is node classification: predicting the primary subject area of each paper. As the setting of OGBN-Arxiv aligns with the Node Attacker 1, we evaluate only Node Attacker 1 on this dataset.
- **Synthetic**: A synthetic dataset containing undirected random graphs with an average degree of 4 and average number of nodes set to 50. The node features are generated by sampling from the standard normal distribution i.e. $\mathcal{N}(0,1)$. Node labels are generated by uniformly sampling an integer between zero and the number of classes specified. Edges are generated by uniformly sampling its two endpoints from the node set. The total number of edges are set to $\frac{d*n}{2}$.

Note that since Node Attacker 2 and Graph Attacker aim at attacking the subgraphs. We randomly sample a node and its 3-hop neighborhood as the subgraph to be attacked. This setting is also consistent with the Federated Graph Learning setting where each client tends to store a subgraph instead of the whole graph.

## 6.1 Node Attacker 1

In this section, we evaluated the performance of feature reconstruction of Node Attacker 1 for the node classification task.

**Target node feature reconstruction.** In this experiment, the attacker has no prior knowledge of the graph dataset. Consequently, a 2-layer tree is generated as a dummy graph, with the target node as the root node and each node having degree $d_{tree} = 10$. Table 3 reports the RNMSE for reconstructing the target node features, where the attack is performed on 20 randomly selected nodes from each dataset. It can be observed that, for GraphSAGE, the attacker achieves high accuracy in reconstructing the nodal features, whereas for GCN, the attacker struggles to reconstruct the target node features reliably. These observations align with the theoretical insights presented in Propositions 1 and 2. Furthermore, Table 3 also shows the minimum RNMSE achieved among the 20 nodes. While the mean RNMSE for GCN may be large, the presence of smaller minimum RNMSE values indicates that leakage of the nodal features can still occur in practice.

A closer examination of our proposed method, GLG, reveals that it generally attains low RNMSE values for GraphSAGE across the Synthetic, FB, OGBN-Arxiv, and GitHub datasets. For example, on Synthetic and FB, GLG reports mean RNMSE values on the order of $10^{-3}$, indicating a near-exact reconstruction of the target node features. Although the mean RNMSE for GitHub is slightly higher, GLG remains highly effective at recovering node features when GraphSAGE is used. Under the GCN framework, while GLG yields larger RNMSE values overall, its minimum RNMSE can still be as low as 0.18 on GitHub. This finding underscores that, even in scenarios where GCN poses a more challenging reconstruction problem, GLG can still reveal critical information about certain nodes.

In comparing GLG and DLG, the results highlight the *superiority of GLG* in most cases. Under the GraphSAGE framework, GLG exhibits significantly lower mean RNMSE on both Synthetic and FB, thereby demonstrating more accurate feature reconstruction. Although DLG achieves a marginally better result on GitHub, this exception does not overshadow the broader trend of GLG's robust performance. Moreover, under the GCN framework, GLG consistently outperforms DLG on all tested datasets.

Detailed results evaluating the performance of node attacker 1 in reconstructing the nodal features of **one-hop neighbors** are presented in Appendix F.1.

## 6.2 Node Attacker 2

In this section, we evaluate the performance of Node Attacker 2 which has access to the gradient of a node classification model for all nodes in a subgraph.

Table 4: Reconstruction of **A** for Node Attacker 2–G.

| Framework | Facebook | | | GitHub | | |
|---|---|---|---|---|---|---|
| | ACC | AUC | AP | ACC | AUC | AP |
| **GLG (Ours)** | | | | | | |
| GraphSAGE | 1.00±0.00 | 1.00±0.00 | 1.00±0.00 | 1.00±0.00 | 1.00±0.00 | 1.00±0.00 |
| GCN | 0.97±0.08 | 0.97±0.02 | 0.85±0.04 | 0.97±0.02 | 0.98±0.001 | 0.87±0.02 |
| **DLG** | | | | | | |
| GraphSAGE | 0.85±0.14 | 0.90± 0.09 | 0.65±0.28 | 0.83±0.16 | 0.87±0.16 | 0.60±0.27 |
| GCN | 0.74±0.25 | 0.74±0.26 | 0.54±0.20 | 0.67±0.28 | 0.71±0.27 | 0.47±0.27 |

Table 5: Reconstruction of **X** for Node Attacker 2–N.

| Framework | Facebook | GitHub |
|---|---|---|
| **GLG (Ours)** | | |
| GraphSAGE ($\times 10^{-4}$) | 0.7±0.6 | 0.9±0.9 |
| GCN | 0.07±0.01 | 0.09±0.02 |
| **DLG** | | |
| GraphSAGE($\times 10^{-3}$) | 0.1±0.0 | 0.06±0.03 |
| GCN | 0.31±0.19 | 0.29±0.24 |

**Node Attacker 2-G.** The reconstruction performance of the adjacency matrix for this setting is listed in Table 4. Note that the attacker **perfectly** reconstructs the graph structure for the GraphSAGE model. For GCN, even though Proposition 3 states that we can exactly reconstruct the adjacency matrix, we do observe some error in the reconstruction. This can be potentially attributed to either the limited number of iterations employed in the attack, hindering its convergence, or the optimization procedure encountering a local minimum. Nonetheless, even though not a perfect reconstruction, the attacker can reconstruct the graph structure with high accuracy and precision for GCN. GLG consistently outperforms DLG for both GraphSAGE and GCN, demonstrating its overall superiority in reconstructing the adjacency matrix.

**Node Attacker 2-N.** Table 5 shows the results for Node Attacker 2-N. The attacker is capable of obtaining almost **perfect** reconstruction of nodal features for the GraphSAGE model. Similarly, for the GCN model, the RNMSE values, although slightly higher, remain sufficiently low, which indicates data leakage. Note that the small error can again be attributed to the potential algorithmic issues such as local optimal as discussed before. Compared to DLG, GLG consistently performs better in reconstructing nodal features for GCN. While DLG achieves slightly better results on the GitHub dataset with GraphSAGE, like Node Attacker 1, this isolated exception does not detract from the overall trend of GLG's reliable performance across settings.

**Node Attacker 2-GN.** In this experimental setting, the attacker possesses no prior information regarding the graph data and endeavors to reconstruct both the nodal features and the graph structure. Table 6 summarizes the corresponding results. For the GraphSAGE model, it is evident that the adversary can reconstruct both the features and the adjacency matrix with high accuracy on both the Facebook and GitHub datasets. These outcomes are consistent with our theoretical findings presented in Proposition 5. Notably, for the GCN framework, the adversary is still able to reconstruct **A** with high AUC and AP scores, despite the absence of a theoretical guarantee for node attacker 2-GN in the GCN context (as noted in Section 5). This empirical observation may be attributed to the integration of the feature smoothness and sparsity regularizers defined in Equation equation 14.

To further evaluate the impact of the regularizers, we assessed the performance of the attack without their inclusion. The corresponding results, also presented in Table 6, indicate a marked decline in the reconstruction performance of the adjacency matrix, particularly in terms of Average Precision (AP), when the regularizers

Table 6: Reconstruction of $\mathbf{X}$ (RNMSE) and $\mathbf{A}$ (AUC and AP) with Node Attacker 2-GN, comparing GLG and DLG.

| Framework | Facebook | | | GitHub | | |
|---|---|---|---|---|---|---|
| | $\mathbf{X}$ (RNMSE) | AUC↑ | AP↑ | $\mathbf{X}$ (RNMSE) | AUC↑ | AP↑ |
| **GLG (Ours)** | | | | | | |
| GraphSAGE | 0.0024±0.003 | 0.99±0.01 | 0.99±0.01 | 0.0024±0.003 | 0.99±0.01 | 0.99±0.01 |
| GCN | 1.39±0.49 | 0.97±0.03 | 0.75±0.10 | 0.58±0.10 | 0.99±0.04 | 0.83±0.04 |
| GCN (w/o reg.) | 1.19±1.08 | 0.95±0.01 | 0.65±0.12 | 0.81±1.41 | 0.94±0.09 | 0.72±0.20 |
| **DLG** | | | | | | |
| GraphSAGE | 0.003±0.0 | 0.99±0.06 | 0.99±0.02 | 0.0026±0.03 | 0.98±0.02 | 0.99±0.1 |
| GCN | 1.01±0.44 | 0.68±0.33 | 0.52±0.38 | 0.88±0.46 | 0.87±0.05 | 0.58±0.18 |

Table 7: Reconstruction of $\mathbf{A}$ with Graph Attacker-G.

| Framework | Facebook | | | GitHub | | |
|---|---|---|---|---|---|---|
| | ACC | AUC | AP | ACC | AUC | AP |
| **GLG (Ours)** | | | | | | |
| GraphSAGE | 1.00±0.00 | 1.00±0.00 | 1.00±0.00 | 1.00±0.00 | 1.00±0.00 | 1.00±0.00 |
| GCN | 0.97±0.03 | 0.93±0.05 | 0.97±0.02 | 0.97±0.02 | 0.95±0.04 | 0.97±0.02 |
| **DLG** | | | | | | |
| GraphSAGE | 0.50±0.01 | 0.50±0.03 | 0.13±0.05 | 0.50±0.04 | 0.51±0.02 | 0.15±0.04 |
| GCN | 0.51±0.01 | 0.51±0.02 | 0.17±0.07 | 0.53±0.01 | 0.53±0.02 | 0.15±0.04 |

are omitted. This suggests that the inclusion of these regularizers effectively enhances the performance of the attack.

Furthermore, we compare the performance of DLG with our proposed method, GLG. Our empirical results demonstrate that DLG is capable of reconstructing both the nodal features and the graph structure only within the GraphSAGE framework, in accordance with Proposition 5. In contrast, for the GCN model on both the Facebook and GitHub datasets, GLG consistently outperforms DLG, particularly with respect to the reconstruction of the adjacency matrix as indicated by the AUC and AP metrics. These findings underscore the broader applicability and superior performance of GLG, which is effective even in scenarios lacking formal theoretical guarantees.

### 6.3 Graph Attacker

In this subsection, the performance of different graph attackers is evaluated for attacking nodal features as well as the adjacency matrix of the private subgraph.

**Graph Attacker-G.** Table 7 lists the reconstruction performance of the graph structure. It can be observed that the attacker can **perfectly** recover the adjacency matrix from the gradients of a GraphSAGE model in both datasets. Such results are also consistent with the results of Proposition 6. On the other hand, in the case of GCN, despite the absence of theoretical guarantees, the attacker demonstrates high accuracy, precision, and AUC values in identifying the edges. Such results can again be attributed to the feature smoothness and sparsity regularizer which help leverage usual properties for social networks. Unlike the Node attacker setting, DLG performs poorly in reconstructing the graph structure under the graph attacker setting, highlighting its limitations in directly handling graph-structured data. To assess the impact of the regularizer, we again conduct the attack with and without the regularizer as shown in equations equation 12 and equation 14. The resulting recovered adjacency matrices are visualized through heatmaps as depicted in Figures 2b and 2c.

Table 8: Reconstruction of **X** (RNMSE) with Graph Attacker-N.

| Framework | Facebook | GitHub |
|---|---|---|
| **GLG (Ours)** | | |
| GraphSAGE | 0.18±0.08 | 0.15±0.08 |
| GCN | 0.61±0.08 | 0.59±0.08 |
| **DLG** | | |
| GraphSAGE | 0.77±0.65 | 0.20±0.4 |
| GCN | 1.32±0.22 | 0.77±0.07 |

Figure 2a illustrates the true graph structure of the private data under attack for comparison. The difference in reconstruction quality between the two scenarios is evident. When the regularizer is not used, the attack accurately identifies a substantial number of edges. However, this also leads to the erroneous identification of many non-edges as edges, resulting in a lower AP score as also shown in the Figure 2.

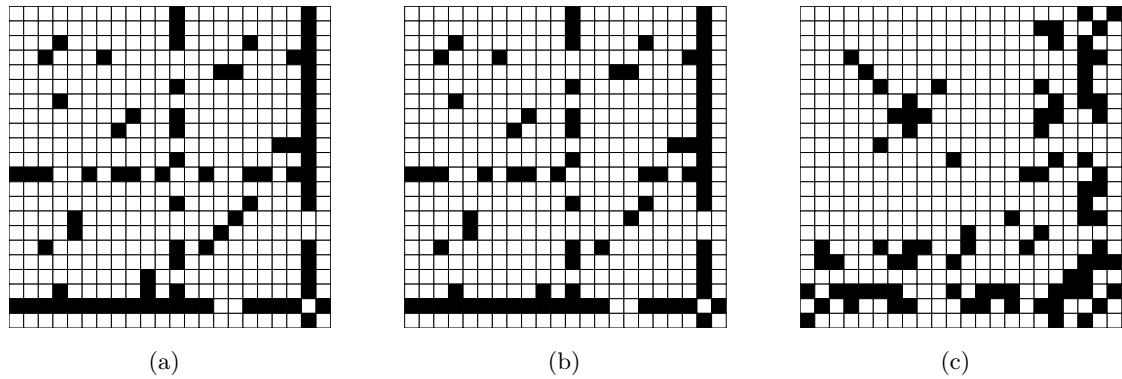

(a)                               (b)                               (c)

Figure 2: The heatmaps of the reconstructed adjacency matrix with Graph Attacker-G from the gradient of GCN models. (a) actual graph. (b) reconstructed graph structure with regularizers (AUC=0.981, AP=1.0). (c) reconstructed graph structure without regularizers (AUC=0.977, AP=0.86).

**Graph Attacker-N.** In this setting, the attacker tries to reconstruct the nodal feature matrix given the gradients and the underlying graph structure. The results are listed in Table 8. It can be observed that while the attacker is unable to reconstruct nodal features from a GCN model, it can recover nodal features from a GraphSAGE model with small errors. The results also corroborate that this attacker is different from Node Attacker 2-N where the gradients of the loss for each node are leveraged to the attacker. Unlike this scenario, Node Attacker 2-N can successfully reconstruct the nodal features. While GLG performs better on GraphSAGE than on GCN, it still consistently outperforms DLG on both architectures, suggesting that DLG struggles to directly handle graph-structured data well due to interdependencies between nodal features and graph topology.

**Graph Attacker-GN.** Table 9 reports the performance of the Graph Attacker-GN in reconstructing the graph structure and nodal features. An analysis of the results for our proposed method, GLG, reveals that the reconstructed adjacency matrix is obtained with high accuracy for the GraphSAGE framework. Interestingly, in this experimental setting, the attack on the GCN framework yields even better performance in recovering the graph structure, as compared to GraphSAGE. It is worth noting, however, that in both frameworks the reconstruction of the nodal features remains unsatisfactory, highlighting an intrinsic difficulty in recovering node attributes under this attack paradigm.

A detailed comparison between GLG and the baseline method, DLG, further underscores the superiority of GLG. In the GraphSAGE setting, while DLG is able to some success reconstruction of the graph structure (for Github), its performance is still far inferior to that of GLG in terms of key metrics such as RNMSE, AUC, and AP. Moreover, even under the GCN framework, GLG exceeds the performance of DLG, despite

Table 9: Reconstruction of **X** (RNMSE) and **A** (AUC and AP) with Graph Attacker-GN for two different methodologies on Facebook and GitHub datasets.

| Framework | Facebook | | | GitHub | | |
|---|---|---|---|---|---|---|
| | $\mathbf{X} \downarrow$ | AUC $\uparrow$ | AP $\uparrow$ | $\mathbf{X} \downarrow$ | AUC $\uparrow$ | AP $\uparrow$ |
| **Ours** | | | | | | |
| GraphSAGE | 0.58±0.10 | 0.89±0.06 | 0.98±0.01 | 0.62±0.09 | 0.92±0.05 | 0.98±0.02 |
| GCN | 0.60±0.09 | 0.92±0.03 | 0.97±0.01 | 0.61±0.03 | 0.93±0.01 | 0.98±0.009 |
| **DLG** | | | | | | |
| GraphSAGE | 1.39±0.01 | 0.50±0.02 | 0.20±0.08 | 0.85±0.73 | 0.71±0.25 | 0.48± 0.46 |
| GCN | 1.31±0.05 | 0.58±0.16 | 0.24±0.14 | 0.78±0.06 | 0.88±0.05 | 0.70±0.08 |

the absence of theoretical guarantees for nodal feature reconstruction. These empirical findings demonstrate that GLG is more robust and effective in adversarial graph reconstruction, providing enhanced accuracy in reconstructing the adjacency matrix even in challenging scenarios.

Overall, the results clearly establish that GLG is a more reliable and superior approach compared to DLG, as it consistently achieves higher reconstruction accuracy across different graph models and datasets.

## 7 Conclusion & Future Work

In this work, we study the gradient inversion attack for Graph Neural Networks (GNNs) in a federated graph learning setting and explore what information can be leaked about the graph data from the gradient. We studied both the node classification and the graph classification tasks, along with two widely used GNN structures, GraphSAGE and GCN. Through both theoretical and empirical results, we highlighted and analyzed the vulnerabilities of different GNN frameworks across a wide variety of gradient inversion attack settings. Our proposed method, GLG, attains superior accuracy in reconstructing both nodal features and graph structure from gradients.

**Future work.** In the future, it would be interesting to study the extension of the attack mechanisms to other GNN frameworks, such as GIN (Graph Isomorphism Networks) (Xu et al., 2019) and GAT (Graph Attention Networks) (Veličković et al., 2018). Experiments and theoretical analysis regarding the vulnerability of other GNN frameworks hence is a promising direction. For example, are certain GNN frameworks more robust to gradient inversion attacks than others? Also, are there any other special properties that can be utilized for graph datasets other than social networks? For instance, it is worth exploiting the fact that the degrees of nodes in molecular datasets exhibit an upper limit. Finally, recent studies such as Wang et al. (2023) have provided theoretical bounds on the reconstruction quality of minibatch data. Hence, an intriguing question would be whether similar guarantees can be established for graph data.

Furthermore, it is crucial to have appropriate defense mechanisms in place to prevent data leakage. It is commonly believed that employing larger batch sizes could be an effective defense against the gradient inversion attack, but our experiments show that even in batches of size 50, the private data can be reconstructed with high accuracy. Therefore, solely relying on larger batch sizes is insufficient as a defense strategy against gradient inversion attacks. Recent studies such as Wen et al. (2022); Wang et al. (2023) have provided provable guarantees regarding the quality of reconstructed private data from training batches. Based on the various defense strategies introduced in Gupta et al. (2022), some potential defense mechanisms can remain effective in graph setting: e.g., noisy gradients (Abadi et al., 2016), gradient pruning (Zhu et al., 2019), encoding inputs (Huang et al., 2020). However, more specific defensive mechanisms from graph perspectives remain undeveloped.

## Acknowledgments

Work in the paper is supported by NSF ECCS 2412484, ECCS 2207457, NSF GEO CI 2425748, and NSF SaTC Frontiers FG22920.

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

## A   Attack Algorithms

---

**Algorithm 1** GLG (Node Attacker).

---

1: **Input:** 2-layer GNN model $F(\mathbf{x}_v, \{\mathbf{x}_{v_j}\}_{j \in \mathcal{N}_v}, \mathbf{W})$
2: $\nabla_{\mathbf{W}}\mathcal{L}_v$: gradients calculated by training data, $\mathbf{y}_v$: True label
3: **Output:** private data $\mathbf{x}_v$
4: //Initialize dummy target and neighboring node features by creating a two level tree with degree as $d_{tree}$ and the label:
5: Initialize $\hat{\mathbf{x}}_v^1$ and $\{\hat{\mathbf{x}}_{v_j}\}_{j \in \mathcal{N}_v}^1$
6: **for** $p \leftarrow 1$ to $P$ **do**
7:   //Compute dummy gradients:
8:   $\nabla_{\mathbf{W}}\hat{\mathcal{L}}_v = \partial\mathcal{L}(F(\hat{\mathbf{x}}_v^p, \{\hat{\mathbf{x}}_{v_j}^p\}_{j \in \mathcal{N}_v}, \mathbf{W}), \mathbf{y}_v)\partial\mathbf{W}$
9:   Compute $\mathbb{D}^p$ as in equation 12
10:   $\hat{\mathbf{x}}_v^{p+1} = \hat{\mathbf{x}}_v^p - \eta\nabla_{\hat{\mathbf{x}}_v^p}\mathbb{D}^p,$
11:   **for** $j \in \mathcal{N}_v$ **do**
12:     $\hat{\mathbf{x}}_{v_j}^{p+1} = \hat{\mathbf{x}}_{v_j}^p - \eta\nabla_{\hat{\mathbf{x}}_{v_j}^p}\mathbb{D}^p,$
13:   **end for**
14: **end for**
15: **return** $\hat{\mathbf{x}} = \hat{\mathbf{x}}_v^{P+1}$

---

**Algorithm 2** GLG (Node Attacker 2)

---

1: **Input:** GNN model $F(\mathbf{X}, \mathbf{A}, \mathbf{W})$, $\mathbf{W}$: parameter weights; $\nabla_{\mathbf{W}}\mathcal{L}$ : gradients calculated by private training data for each node in the graph, $\mathbf{y}$: True label
2: **Output:** private training data $\mathbf{X}, \mathbf{A}$
3: Initialize dummy features and labels $\hat{\mathbf{X}}^1, \hat{\mathbf{A}}^1$
4: **for** $p \leftarrow 1$ to $P$ **do**
5:   //Compute dummy gradients:
6:   $\nabla_{\mathbf{W}}\hat{\mathcal{L}} = \partial\mathcal{L}(F(\hat{\mathbf{X}}^p, \hat{\mathbf{A}}^p, \mathbf{W}), \mathbf{y})/\partial\mathbf{W}$
7:   Compute $\hat{\mathbb{D}}^p$ as given in equation 14
8:   $\hat{\mathbf{X}}^{p+1} = \hat{\mathbf{X}}^p - \eta\nabla_{\hat{\mathbf{X}}^p}\hat{\mathbb{D}}^p,$
9:   $\hat{\mathbf{A}}^{p+1} = \text{proj}_{[0,1]}(\hat{\mathbf{A}}^p - \eta\nabla_{\hat{\mathbf{A}}^p}\hat{\mathbb{D}}^p),$
10: **end for**
11: **return** $\hat{\mathbf{X}} = \hat{\mathbf{X}}^{P+1}, \hat{\mathbf{A}} \sim Ber(\mathbf{A}^{P+1})$

---

**Algorithm 3** Graph Attacker

---

1: **Input:** GNN model $F(\mathbf{X}, \mathbf{A}, \mathbf{W})$, $\mathbf{W}$: parameter weights; $\nabla_{\mathbf{W}}\mathcal{L}$: gradients calculated by private training data, $\mathbf{y}$: True label
2: **Output:** private training data $\mathbf{X}, \mathbf{A}$
3: Initialize dummy features and labels $\hat{\mathbf{X}}^1, \hat{\mathbf{A}}^1$
4: **for** $p \leftarrow 1$ to $P$ **do**
5:   //Compute dummy gradients:
6:   $\nabla_{\mathbf{W}}\hat{\mathcal{L}} = \partial\mathcal{L}(F(\hat{\mathbf{X}}^p, \hat{\mathbf{A}}^p, \mathbf{W}), \mathbf{y})/\partial\mathbf{W}$
7:   Compute $D^p$ as given in equation 12
8:   $\hat{\mathbf{X}}^{p+1} = \hat{\mathbf{X}}^p - \eta\nabla_{\hat{\mathbf{X}}^p}\mathbb{D}^p,$
9:   $\hat{\mathbf{A}}^{p+1} = \text{proj}_{[0,1]}(\hat{\mathbf{A}}^p - \eta\nabla_{\hat{\mathbf{A}}^p}\mathbb{D}^p),$
10: **end for**
11: **return** $\hat{\mathbf{X}} = \hat{\mathbf{X}}^{P+1}, \hat{\mathbf{A}} \sim Ber(\mathbf{A}^{P+1})$

---

## B  Extracting ground-truth labels

In a classification task, it is possible to extract the ground-truth labels from the gradients as shown in Zhao et al. (2020a). Similary, it is possible to extract the ground-truth labels in a graph/node classification task. We demonstrate how to extract the ground-truth labels when cross-entropy loss is used as defined in equation 5. The gradients of the loss with respect to $\mathbf{p}_i$ is given by

$$g_i = \frac{\partial \mathcal{L}}{\partial \mathbf{p}_i} = \begin{cases} -1 + \left( \frac{e^{\mathbf{P}_i}}{\sum_j e^{\mathbf{P}_j}} \right) & \text{if } i = k, \\ \left( \frac{e^{\mathbf{P}_i}}{\sum_j e^{\mathbf{P}_j}} \right) & \text{else.} \end{cases} \tag{22}$$

Since $\left( \frac{e^{\mathbf{P}_i}}{\sum_j e^{\mathbf{P}_j}} \right) \in (0,1)$ therefore $g_i \in (-1,0)$ when $i = k$ and $g_i \in (0,1)$ otherwise. Now consider the node classification task for a GCN layer as defined in equation 3. For the final layer of the model with output as $\mathbf{h}_v^L = \mathbf{p}$ we can write the gradients of the loss as follows

$$\nabla_{(\mathbf{w})_i} \mathcal{L} = \frac{\partial \mathcal{L}}{\partial (\mathbf{W})_i} = \frac{\partial \mathcal{L}}{\partial \mathbf{p}_i} \frac{\partial \mathbf{p}_i}{\partial (\mathbf{W})_i} \tag{23}$$

$$= g_i \cdot \frac{\partial \left( \sigma \left( \mathbf{h}_v^{\text{agg}}(\mathbf{W})_i^\top + \mathbf{b}_i \right) \right)}{\partial (\mathbf{W})_i} \tag{24}$$

$$= g_i \cdot \lambda_i \cdot \mathbf{h}_v^{\text{agg}}. \tag{25}$$

As $\mathbf{h}_v^{\text{agg}}$ is independent of the logit index $i$, and $\lambda_i \geq 0$ because of the sigmoid non-linearity, the ground-truth label $k$ can be identified by just checking the signs of $\nabla_{(\mathbf{w})_i} \mathcal{L}$ for all $i$. Formally, the ground-truth label $k$ is the one for which the following condition holds

$$k = i \text{ s.t. } \nabla_{(\mathbf{w})_i} \mathcal{L} \cdot \nabla_{(\mathbf{w})_j} \mathcal{L} \leq 0, \ \forall i \neq j. \tag{26}$$

## C  Proof of Proposition 1 for GraphSAGE

For GraphSAGE layer defined in equation 1 the gradient equations for the first layer can be written as follows

$$\mathbf{h}_v = \sigma(\tilde{\mathbf{h}}_v), \tag{27}$$

$$\tilde{\mathbf{h}}_v = \mathbf{x}_v^{\text{agg}} \mathbf{W}_1^\top + \mathbf{x}_v \mathbf{W}_2^\top + \mathbf{b}, \tag{28}$$

$$\nabla_{(\mathbf{W}_1)_i} \mathcal{L}_v = \nabla_{(\tilde{\mathbf{h}}_v)_i} \mathcal{L}_v \cdot \nabla_{(\mathbf{W}_1)_i} (\tilde{\mathbf{h}}_v)_i, \tag{29}$$

$$\nabla_{(\mathbf{W}_1)_i} \mathcal{L}_v = \nabla_{(\mathbf{b})_i} \mathcal{L}_v \cdot \mathbf{x}_v^{\text{agg}} \tag{30}$$

## D  Proof of Proposition 2

For a GraphSAGE layer as shown in equation 1 the target nodal features can be recovered as the following

$$\mathbf{h}_v = \sigma(\tilde{\mathbf{h}}_v), \tag{31}$$

$$\tilde{\mathbf{h}}_v = \mathbf{x}_v^{\text{agg}} \mathbf{W}_1^\top + \mathbf{x}_v \mathbf{W}_2^\top + \mathbf{b}, \tag{32}$$

$$\nabla_{(\mathbf{W}_2)_i} \mathcal{L}_v = \nabla_{(\tilde{\mathbf{h}}_v)_i} \mathcal{L}_v \cdot \nabla_{(\mathbf{W}_2)_i} (\tilde{\mathbf{h}}_v)_i, \tag{33}$$

$$\nabla_{(\mathbf{W}_2)_i} \mathcal{L}_v = \nabla_{(\mathbf{b})_i} \mathcal{L}_v \cdot \mathbf{x}_v. \tag{34}$$

The target nodal features can henceforth be reconstructed as $\mathbf{x}_v = \nabla_{(\mathbf{W}_2)_i} \mathcal{L}_v / \nabla_{(\mathbf{b})_i} \mathcal{L}_v$ as long as $\nabla_{(\mathbf{b})_i} \mathcal{L}_v \neq 0$.

## E   Proof of Proposition 6

*Proof.* For a GraphSAGE framework as shown in equation 6, let $\tilde{\mathbf{H}}$ be the hidden node representation matrix before applying the non-linearity as shown below

$$\mathbf{H} = \sigma(\tilde{\mathbf{H}}), \tag{35}$$

$$\tilde{\mathbf{H}} = \tilde{\mathbf{A}}\mathbf{X}\mathbf{W}_1^\top + \mathbf{X}\mathbf{W}_2^\top + \overrightarrow{\mathbf{1}}\mathbf{b}. \tag{36}$$

The derivatives of the loss function with respect to the weights $\mathbf{W}_1, \mathbf{W}_2$, and bias $\mathbf{b}$ of the first convolutional layer can be written as

$$\frac{\partial \mathcal{L}}{\partial (\mathbf{W}_1)_i} = \sum_{k=1}^{N} \frac{\partial \mathcal{L}}{\partial \tilde{\mathbf{H}}_{ki}} \frac{\partial \tilde{\mathbf{H}}_{ki}}{\partial (\mathbf{W}_1)_i} = \sum_{k=1}^{N} \frac{\partial \mathcal{L}}{\partial \tilde{\mathbf{H}}_{ki}} (\tilde{\mathbf{A}}\mathbf{X})_k, \tag{37}$$

$$\frac{\partial \mathcal{L}}{\partial (\mathbf{W}_2)_i} = \sum_{k=1}^{N} \frac{\partial \mathcal{L}}{\partial \tilde{\mathbf{H}}_{ki}} (\mathbf{X})_k, \tag{38}$$

$$\frac{\partial \mathcal{L}}{\partial (\mathbf{b})_i} = \sum_{k=1}^{N} \frac{\partial \mathcal{L}}{\partial \tilde{\mathbf{H}}_{ki}}, \tag{39}$$

where $\partial$ denotes the partial derivative operator. Equations equation 37-equation 39 can be written in matrix form as the following

$$(\tilde{\mathbf{A}}\mathbf{X})^\top \nabla_{\tilde{\mathbf{H}}}\mathcal{L} = \nabla_{\mathbf{W}_1}\mathcal{L}^\top, \tag{40}$$

$$\mathbf{X}^\top \nabla_{\tilde{\mathbf{H}}}\mathcal{L} = \nabla_{\mathbf{W}_2}\mathcal{L}^\top, \tag{41}$$

$$\nabla_{\tilde{\mathbf{H}}}\mathcal{L}^\top \cdot \overrightarrow{\mathbf{1}} = \nabla_{\mathbf{b}}\mathcal{L}^\top. \tag{42}$$

It can be observed from equation 40-equation 41 that there are two unknowns $\tilde{\mathbf{A}}$ and $\nabla_{\tilde{\mathbf{H}}}\mathcal{L}$. However $\tilde{\mathbf{A}}$ and $\nabla_{\tilde{\mathbf{H}}}\mathcal{L}$ can be computed directly from the above equations. Given equation 41 and the fact that $\mathbf{X}$ is full-row rank, $\nabla_{\tilde{\mathbf{H}}}\mathcal{L}$ can be obtained as follows

$$\nabla_{\tilde{\mathbf{H}}}\mathcal{L} = (\mathbf{X}^\top)^\dagger \nabla_{\mathbf{W}_2}\mathcal{L}^\top. \tag{43}$$

Plugging the value of $\nabla_{\tilde{\mathbf{H}}}\mathcal{L}$ in equation 40, $\tilde{\mathbf{A}}$ can be calculated as

$$\tilde{\mathbf{A}} = \mathbf{X}(\nabla_{\mathbf{W}_2}\mathcal{L})^\dagger (\nabla_{\mathbf{W}_1}\mathcal{L})\mathbf{X}^\dagger \tag{44}$$

.                                                                                                    □

## F   Supplementary Experimental Results and Analysis

Table 10: Dataset statistics for Node Classification task.

| Dataset | #Nodes | #Edges | #Labels |
|---------|--------|--------|---------|
| GitHub | 37,300 | 578,006 | 2 |
| Facebook | 22,470 | 342,004 | 4 |

### F.1   Node Attacker 1: One-hop Neighborhood Nodal Feature Reconstruction

To evaluate the performance of the node attacker in reconstructing the nodal features of one-hop neighbors, a target node is selected randomly from the graph. A dummy graph is initialized as a depth-2 tree with $d_{tree}$ set to the actual number of one-hop neighbors of the target node. Since there's permutation ambiguity in the reconstructed nodal features, the Hungarian Algorithm is used to find the best match with the original

Table 11: Target node feature reconstruction $\mathbf{x}_v$ (RNMSE $\times 10^{-2}$) for Node Attacker 1 in a batched setting.

| Dataset | | $B=5$ | $B=20$ | $B=50$ |
|---|---|---|---|---|
| Synth. | Mean | 0.07±0.02 | 0.8±0.3 | 3.9±1.4 |
| | Min | 0.0 | 0.13 | 0.4 |
| FB | Mean | 0.9±0.4 | 12±64 | 17±95 |
| | Min | 0.8 | 28 | 65 |
| Github | Mean | 28±14 | 126±78 | 132±83 |
| | Min | 0.3 | 111 | 107 |

Table 12: Reconstruction performance of $\mathbf{X}$ (RNMSE $\times 10^{-2}$) for Graph Attacker-N for batched graph dataset.

| $d$ | | $B=5$ | $B=20$ | $B=50$ |
|---|---|---|---|---|
| 4 | Mean | 23 ± 31 | 82 ± 27 | 119±1 |
| | Min | 0.04 | 9 | 86 |
| 8 | Mean | 25±33 | 84±34 | 121±9 |
| | Min | 0.04 | 9 | 87 |
| 12 | Mean | 22±30 | 89±36 | 121±9 |
| | Min | 0.04 | 9 | 87 |

neighboring node features (Kuhn, 2010). Figure 3 shows the one-hop neighbor reconstruction error (RNMSE) using GraphSAGE. The results are averaged over 10 randomly selected nodes. For some nodes, the nodal features of neighbors were reconstructed accurately. For example, nodes 0 and 8 in Figure 3a for Facebook. Further, although the mean RNMSE of the reconstructed neighbors is high, the minimum RNMSE of the best-reconstructed neighbors can be low (node 9 in Figure 3a), indicating private data leakage.

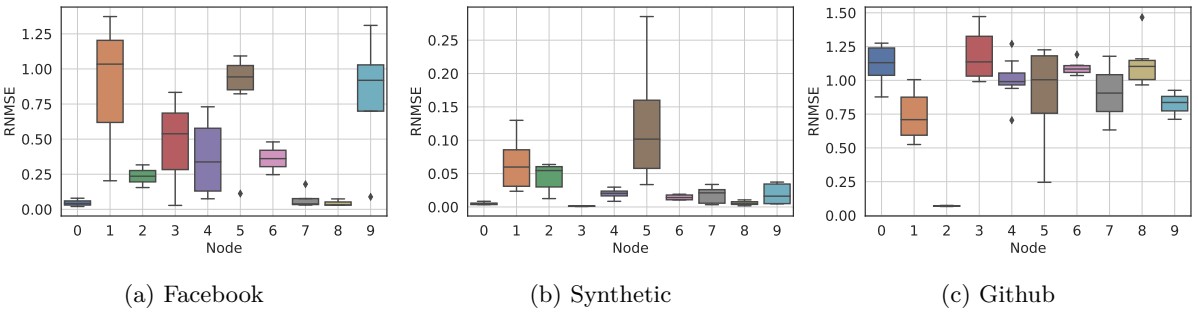

(a) Facebook      (b) Synthetic      (c) Github

Figure 3: One-hop neighbor reconstruction (RNMSE) for Node Attacker-1 with GraphSAGE.

## F.2   Attacking batched data

We also extended the experiments for node and graph attackers to the minibatch setting. Specifically, we only evaluate GraphSAGE for both tasks since it provides the best reconstruction in the stochastic (single data sample) setting as shown in the previous experiments. Although effective attack techniques (Wen et al., 2022) have been proposed to reconstruct image data from large batches, here we tested the performance of the attackers by simply optimizing with respect to the dummy input batch. Hence, the performance may improve if attack techniques for batch data are also incorporated. Due to possible permutation ambiguity in a batch setting, the Hungarian algorithm is used to find the best match with the actual input data. The evaluation metric for the algorithm is the RNMSE between all pairs of the dummy batch and the actual batch. These results are averaged over 10 independent runs for each batch size. Experiments are conducted for Node Attacker 1 and Graph Attacker-N.

**Node Attacker 1.** Table 11 shows the performance of reconstructing the target node features for batches of size $B = 5, 20$, and 50. It can be observed that the best results are obtained for the Synthetic Dataset with accurate reconstruction across all batch sizes. When the batch size is $B = 5$, the attack provides accurate reconstruction across all the 3 datasets.

**Graph Attacker-N.** In this scenario, the attack attempts to reconstruct $\mathbf{X}$ for the whole batch with $\mathbf{A}$ being known. To simplify the matching after running the attack we only consider graphs with the same number of nodes. This is done by generating Erdos-Renyi (ER) graphs with the number of nodes $n = 50$. For a ER graph with $n$ nodes and edge probability $p$, the average degree of a node is $d = (n-1)p$. We conduct experiments by varying $d$ with different batch sizes. The nodal features are generated by sampling from the standard normal distribution i.e. $\mathcal{N}(0, 1)$. Table 12 lists the performance of reconstructing $\mathbf{X}$ with varying batch sizes and degrees. As expected, the quality of reconstruction declines with increasing batch size, with the best results being achieved at $B = 5$. Also, the performance remains roughly the same across all degrees.

### F.3 Sensitivity Analysis for Hyperparameters

In our attack, numerous hyperparameters come into play, including but not limited to the regularization parameters $\alpha$ and $\beta$, the graph size, and the network width. In our analysis, we examined a subset of these hyperparameters in different settings and subsequently present the corresponding outcomes below.

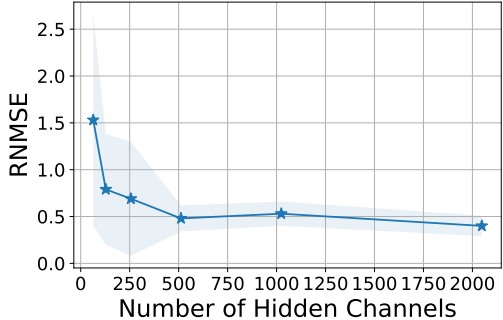

Figure 4: Variation in RNMSE of the reconstructed target features vs. the number of hidden channels for a GCN framework on the Github dataset.

**Network Width.** All previous experiments were conducted with the number of hidden channels set to 100. In the setting of Node Attacker 1, the attack could *successfully* reconstruct the target nodal features for a GraphSAGE framework with high accuracy. However, for a GCN framework, the attack only managed to reconstruct the aggregated nodal features. Consequently, in this section, we explore the impact of varying the number of hidden channels in the 2-layer GCN framework on the quality of reconstruction. Figure 4 illustrates the results of target node feature reconstruction for the Github Dataset. Notably, as the number of hidden channels increases, there is a clear decrease in the RNMSE values, suggesting that wider networks are more susceptible to the attack.

**Regularization Hyperparameters.** To investigate the influence of $\alpha$ and $\beta$ in equation 14, we examine the effects on the reconstruction performance of Graph Attacker-G for the GCN framework. More specifically, we maintain one hyperparameter at a fixed value while altering the other, and depict the AUC and AP values. Figure 5a illustrates the AUC and AP values for different $\alpha$ values in a log scale with base 10. The values on the x-axis are also in the log scale with base 10. In Figure 5a, $\alpha$ is varied while keeping $\beta = 10^{-7}$. Similarly, in Figure 5b, where $\beta$ is varied, $\alpha$ is fixed at $10^{-9}$. It is evident from the figures that higher values of hyperparameters severely degrade the performance of the attack whereas extremely low values give suboptimal results. This clearly demonstrates the effectiveness of the feature smoothness and sparsity regularizers.

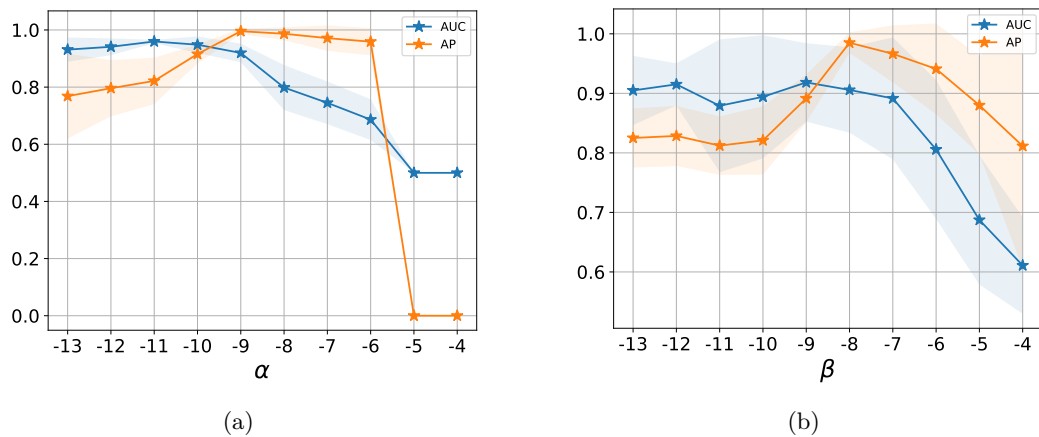

Figure 5: Variation of AUC and AP vs. $\alpha$ and $\beta$ for Graph Attacker-G.

## G  Additional Graph Classification results on TUDatasets

All attack settings considered in previous experiments in Section 6 are defined within the context of social networks. In order to assess the efficacy of the attack on alternative datasets, such as molecular datasets which are commonly employed in graph classification tasks, we extend our analysis to the TUDataset (Morris et al., 2020).

For the graph classification task, we conduct the attack on the following three datasets from the TUDataset repository (Morris et al., 2020).

- **MUTAG** (Debnath et al., 1991): It consists of 188 chemical compounds from two classes. The vertices represent atoms and edges represent chemical bonds. Node features denote the atom type represented by one-hot encoding. Labels represent their mutagenic effect on a specific gram-negative bacterium.

- **COIL-RAG** (Riesen & Bunke, 2008): Dataset in computer vision where images of objects are represented as region adjacency graphs. Each node in the graph is a specific part in the original image. Graph labels denote the object type.

- **FRANKENSTEIN** (Kazius et al., 2005): It contains graphs representing chemical molecules, where the vertices are chemical atoms and edges represent the bonding. Node features denote the atom type. Binary labels represent mutagenecity.

The average number of nodes in the graphs of these datasets is significantly smaller than social networks (see Table 13). However, unlike social networks, these graphs might not necessarily exhibit properties like feature smoothness and sparsity. Hence, for all the attacks we use the objective function as defined in equation 12. The results in all attack settings are again consistent with the theoretical analysis.

The results are obtained by randomly selecting 20 graphs from the datasets. The selected graphs are fixed for different parameter selections for fair comparison. In this case since the graphs are small we can directly use the Mean Absolute Error (MAE) of the reconstructed adjacency matrix to measure the performance of reconstructing the underlying graph structure. Specifically, the MAE is given as follows

$$\text{MAE}(\mathbf{A}, \hat{\mathbf{A}}) = \frac{2 * |\text{tri}_{\text{lower}}(\mathbf{A}) - \text{tri}_{\text{lower}}(\hat{\mathbf{A}})|}{N(N+1)} \tag{45}$$

Also, instead of directly sampling edges from the probabilistic adjacency matrix as given in Section 4.2, the MAE is also evaluated with and without thresholding the reconstructed adjacency matrix. We apply min-max normalization to the resulting $\hat{\mathbf{A}}$ before thresholding. After applying the threshold, $\hat{\mathbf{A}}_\tau := th(\hat{\mathbf{A}}, \tau)$ is obtained, where $th(\mathbf{Z}, \tau)$ is a entry-wise operator that returns 0 if $z_{ij} < \tau$, and returns 1 otherwise.

Table 13: Dataset statistics (Graph Classification). Edge density is calculated based on (#Avg.Edges/#Edges in the fully connected graph).

| Dataset | #Avg. Nodes | #Avg. Edge | #Edge Density | #Labels |
|---|---|---|---|---|
| MUTAG | 17.93 | 19.79 | 0.1304 | 2 |
| COIL-RAG | 3.01 | 3.02 | 0.9983 | 100 |
| FRANK. | 16.90 | 17.88 | 0.1331 | 2 |

Table 14: Reconstruction of $\mathbf{A}$ (MAE) for Graph Attacker-G

(a) GraphSAGE

| $\tau$ | MUTAG | COIL-RAG | FRANK.($\times 10^{-4}$) |
|---|---|---|---|
| N/A | 0.25±0.10 | 0.02±0.04 | 0.08±0.3 |
| 0.2 | 0.27±0.11 | 0.03±0.07 | 0.0±0.0 |
| 0.4 | 0.24±0.11 | 0.03±0.06 | 0.0±0.0 |
| 0.6 | 0.23±0.11 | 0.02 ± 0.04 | 0.0±0.0 |
| 0.8 | 0.23±0.10 | 0.003±0.13 | 0.0±0.0 |

(b) GCN

| $\tau$ | MUTAG | COIL-RAG | FRANK. ($\times 10^{-2}$) |
|---|---|---|---|
| N/A | 0.20±0.04 | 0.22±0.11 | 4.58±2.59 |
| 0.2 | 0.27±0.07 | 0.16±0.09 | 4.74±3.3 |
| 0.4 | 0.19±0.06 | 0.16±0.09 | 4.20±2.15 |
| 0.6 | 0.15±0.04 | 0.16±0.09 | 4.08±1.96 |
| 0.8 | 0.14±0.01 | 0.26±0.267 | 4.22±2.04 |

**Graph Attacker-G.** Table 14 shows the MAE of reconstructing $\mathbf{A}$ with thresholding (for different values of $\tau$) and without (N/A in the table). The attack achieves the best results on the FRANKENSTEIN dataset while using GraphSAGE. Each node in the FRANKENSTEIN dataset has $D = 780$ features, whereas MUTAG and COIL-RAG have only $D = 7$ and $D = 64$ features respectively. Moreover, the average number of nodes ($N$) per graph in FRANKENSTEIN, MUTAG and COIL-RAG are 16.9, 17.9 and 3.01 respectively. This implies that $\mathbf{X} \in \mathbb{R}^{N \times D}$ is full-row rank with high probability in COIL-RAG and FRANKENSTEIN. Meanwhile, the node features in the MUTAG dataset consist of only binary values. It can be observed from the experimental results that the best and worst results are obtained on FRANKENSTEIN and MUTAG respectively. This is also consistent with Proposition 6 which requires $\mathbf{X}$ to be full row-rank for successfully reconstructing $\tilde{\mathbf{A}}$. The reconstruction performance improves for COIL-RAG with GraphSAGE at $\tau = 0.8$. The FRANKENSTEIN dataset is reconstructed accurately even with the GCN framework.

**Graph Attacker-N.** Table 15 lists the RNMSE of reconstructing $\mathbf{X}$. It can be observed that the attack is able to reconstruct $\mathbf{X}$ with high accuracy for GraphSAGE across all three datasets. For the GCN framework, it can be observed that the algorithm fails to reconstruct $\mathbf{X}$.

**Graph Attacker-GN.** In this scenario, since the attacker has no knowledge of the graph data, we try different settings and observe their effects on the reconstruction. To improve the attack for Graph Attacker-GN where we have no prior knowledge of the graph, we utilize various initialization and thresholding strategies. Specifically, since the optimization process given in the algorithms can be greatly improved by utilizing different starting points, we initialize the dummy node feature and adjacency matrices with different values. In the case of thresholding, instead of sampling the edges given the probabilistic adjacency matrix $\hat{\mathbf{A}}^{P+1}$, we use various threshold values to sample edges.

**The effect of initialization:** Previous studies have established that the quality of reconstruction heavily depends on the initialization (Gupta et al., 2022; Geiping et al., 2020). The closer the initialization is to the original data, the better the reconstruction. An initialization value of 0.1 implies that all the entries of $\hat{\mathbf{X}}$ and $\hat{\mathbf{A}}$ are initialized to 0.1 and similarly $\mathcal{N}(0, 1)$ implies that all entries are initialized from the standard

Table 15: Reconstruction of $\mathbf{X}$ (RNMSE) for Graph Attacker-N.

| | MUTAG | COIL-RAG | FRANK. |
|---|---|---|---|
| GraphSAGE($\times 10^{-2}$) | 0.07±0.06 | 6.91±7.1 | 21.95±16.05 |
| GCN | 8.80±1.27 | 1.51±0.17 | 2.68±0.30 |

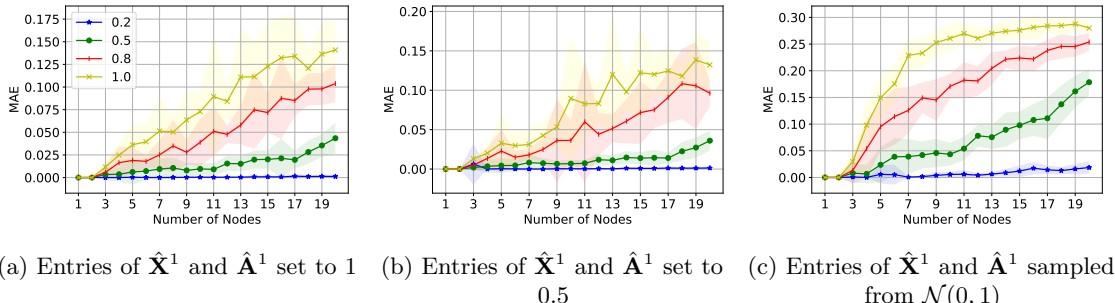

(a) Entries of $\hat{\mathbf{X}}^1$ and $\hat{\mathbf{A}}^1$ set to 1    (b) Entries of $\hat{\mathbf{X}}^1$ and $\hat{\mathbf{A}}^1$ set to 0.5    (c) Entries of $\hat{\mathbf{X}}^1$ and $\hat{\mathbf{A}}^1$ sampled from $\mathcal{N}(0,1)$

Figure 6: Reconstruction of different graph structures (MAE) for Graph Attacker-GN using GraphSAGE.

Table 16: Reconstruction of $\mathbf{A}$ (MAE) in for Graph Attacker-GN with different thresholds.

(a) GraphSAGE

| $\tau$ | MUTAG | COIL-RAG | FRANK. |
|---|---|---|---|
| N/A | 0.37±0.10 | 0.18±0.11 | 0.41±0.17 |
| 0.2 | 0.47 ±0.13 | 0.20±0.11 | 0.64±0.25 |
| 0.4 | 0.43±0.13 | 0.15±0.14 | 0.50±0.24 |
| 0.5 | 0.39±0.13 | 0.14±0.17 | 0.38±0.20 |
| 0.6 | 0.32±0.08 | 0.11±0.18 | 0.29±0.15 |
| 0.8 | 0.26±0.06 | 0.19±0.18 | 0.19±0.10 |

(b) GCN

| $\tau$ | MUTAG | COIL-RAG | FRANK. |
|---|---|---|---|
| N/A | 0.48±0.05 | 0.20±0.12 | 0.41±0.08 |
| 0.2 | 0.68 ±0.06 | 0.15±0.08 | 0.64±0.08 |
| 0.4 | 0.55 ±0.07 | 0.14±0.08 | 0.46±0.12 |
| 0.5 | 0.47 ±0.07 | 0.14±0.08 | 0.38±0.11 |
| 0.6 | 0.40±0.07 | 0.14±0.08 | 0.32±0.10 |
| 0.8 | 0.30±0.05 | 0.32±0.34 | 0.19±0.08 |

normal distribution. Since the values of $\mathbf{X}$ and $\mathbf{A}$ lie in the range $[0, 1]$ for all three datasets, we tested the effect of different initialization points in the same range. Specifically, we tested the initialization values as $\{0, 0.1, 0.5, 1, \mathcal{N}(0, 1)\}$. The results of the attack with different initialization strategies is shown in Table 18. The leftmost column gives the initialization values of the dummy adjacency and node feature matrices. For the adjacency matrix, the results are listed with and without thresholding. Similar to Scenario 2, we apply min-max normalization (before thresholding) and a threshold value of 0.5 is chosen. A more compact version of the results showing the best performance over all initialization strategies is shown in Table 17. It can be observed that for GraphSAGE, the attack can accurately reconstruct both $\mathbf{X}$ and $\mathbf{A}$ in the MUTAG and FRANKENSTEIN datasets. While using GCN, even though $\mathbf{X}$ cannot be recovered, $\mathbf{A}$ can be reconstructed accurately.

For GraphSAGE in Table 18a, it can be observed that even without any knowledge of the graph data, the attack can accurately reconstruct both $\mathbf{X}$ and $\mathbf{A}$ just by utilising a good initialization strategy. For example, in Table 18a for the FRANKENSTEIN dataset, the attack gives a high error for both $\mathbf{A}$ and $\mathbf{X}$ while using $\mathcal{N}(0, 1)$ as the initialization. However, the MAE drops by orders of magnitude when we initialize $\mathbf{X}$ and $\mathbf{A}$ to a value of 0.2. Even for the GCN framework in Table 18b, while the data cannot be reconstructed in most of the cases, it can be observed that a good initialization point can leak $\mathbf{A}$ in COIL-RAG and FRANKENSTEIN datasets.

**The effect of threshold:** To explore the effect of thresholding on the adjacency matrix, we examined the MAE across all three datasets using different threshold values. Table 16 lists the MAE of reconstructing $\mathbf{A}$ at different thresholds. It can be observed that unlike Scenario 2, the performance of the algorithm stays roughly the same for both GraphSAGE and GCN. Also, the FRANKENSTEIN dataset is no longer vulnerable to the attack unlike Scenario 2. Few of the threshold values reduce the MAE by a considerable amount. For example, $\tau = 0.8$ works well for MUTAG and FRANKENSTEIN in the case of GraphSAGE, and for MUTAG in the case of GCN.

**Vulnerable graph structures:** In order to identify vulnerable graph structures, we tested the performance of the attack in attacking graphs with varying numbers of nodes and edges for the GraphSAGE framework.

Table 17: Reconstruction of $\mathbf{X}$ (RNMSE) and $\mathbf{A}$ (MAE) for Graph Attacker-GN.

| Dataset | GraphSAGE | | | GCN | | |
|---|---|---|---|---|---|---|
| | $\hat{\mathbf{X}}$ | $\hat{\mathbf{A}}$ | $\hat{\mathbf{A}}_{0.5}$ | $\hat{\mathbf{X}}$ | $\hat{\mathbf{A}}$ | $\hat{\mathbf{A}}_{0.5}$ |
| MUTAG | 0.0027±0.0078 | 0.22±0.09 | 0.21±0.11 | 1.07±0.20 | 0.24±0.02 | 0.14±0.04 |
| COIL-RAG | 0.005 ± 0.0015 | 0.01±0.03 | 0.001±0.006 | 0.74±0.13 | 0.05±0.10 | 0.03±0.10 |
| FRANK. | 0.01±0.006 | 0.004 ± 0.002 | 0.00±0.00 | 0.63±0.18 | 0.07±0.02 | 0.007±0.01 |

Table 18: Reconstruction of $\mathbf{X}$ (RNMSE) and $\mathbf{A}$ (MAE) for Graph Attacker-GN.

(a) GraphSAGE

| init | MUTAG | | | COIL-RAG ($\times 10^{-2}$) | | | FRANK. ($\times 10^{-2}$) | | |
|---|---|---|---|---|---|---|---|---|---|
| | $\hat{\mathbf{X}}(\times 10^{-2})$ | $\hat{\mathbf{A}}$ | $\hat{\mathbf{A}}_{0.5}$ | $\hat{\mathbf{X}}$ | $\hat{\mathbf{A}}$ | $\hat{\mathbf{A}}_{0.5}$ | $\hat{\mathbf{X}}$ | $\hat{\mathbf{A}}$ | $\hat{\mathbf{A}}_{0.5}$ |
| 0.0 | 4±1 | 0.37±0.10 | 0.47±0.13 | 31±8 | 1±9 | 4±8 | 29±14 | 27±15 | 21±15 |
| 0.1 | 0.9±0.9 | 0.23±0.08 | 0.26 ± 0.14 | 22±1 | 9±8 | 3±7 | 4±5 | 1±2 | 0.6±1 |
| 0.2 | 0.8±2 | 0.22±0.09 | 0.21±0.11 | 9±5 | 5±4 | 0.5±2 | 1±0.6 | 0.4±0.2 | 0.0±0.0 |
| 0.5 | 0.3±0.4 | 0.23±0.10 | 0.22±0.12 | 0.6±1 | 1±3 | 0.1±0.6 | 1±0.9 | 0.4±0.3 | 0.0±0.0 |
| 1.0 | 0.2±0.7 | 0.23±0.10 | 0.22±0.11 | 0.5±1 | 1±3 | 0.1±0.6 | 1±4 | 0.4 ±0.8 | 0.06±0.2 |
| $\mathcal{N}(0,1)$ | 6 ± 3 | 0.37 ± 0.10 | 0.39 ± 0.13 | 26 ± 12 | 18 ± 11 | 14 ± 17 | 66 ± 43 | 41 ± 17 | 38 ± 20 |

(b) GCN

| init | MUTAG | | | COIL-RAG | | | FRANK. | | |
|---|---|---|---|---|---|---|---|---|---|
| | $\hat{\mathbf{X}}$ | $\hat{\mathbf{A}}$ | $\hat{\mathbf{A}}_{0.5}$ | $\hat{\mathbf{X}}$ | $\hat{\mathbf{A}}$ | $\hat{\mathbf{A}}_{0.5}$ | $\hat{\mathbf{X}}$ | $\hat{\mathbf{A}}$ | $\hat{\mathbf{A}}_{0.5}$ |
| 0.0 | 1.38±0.31 | 0.25±0.05 | 0.21±0.07 | 1.96±2.2 | 0.05±0.10 | 0.03±0.10 | 1.11±0.27 | 0.11±0.03 | 0.03±0.04 |
| 0.1 | 1.06 ± 0.15 | 0.24±0.02 | 0.14±0.04 | 1.54±0.1.48 | 0.43±0.44 | 0.42±0.47 | 0.66±0.15 | 0.07±0.02 | 0.007±0.01 |
| 0.2 | 1.07±0.20 | 0.25 ± 0.02 | 0.14±0.04 | 1.12±0.71 | 0.50±0.40 | 0.47±0.43 | 0.63±0.18 | 0.08±0.02 | 0.007±0.01 |
| 0.5 | 1.47±0.43 | 0.33±0.06 | 0.23±0.09 | 0.75±0.14 | 0.17±0.33 | 0.17±0.35 | 0.72±0.25 | 0.14±0.05 | 0.05±0.07 |
| 1.0 | 1.15±0.33 | 0.34±0.04 | 0.26±0.06 | 0.74±0.13 | 0.76±0.31 | 0.78±0.32 | 0.81±0.23 | 0.19±0.05 | 0.13±0.06 |
| $\mathcal{N}(0,1)$ | 3.26±0.36 | 0.48 ± 0.05 | 0.47 ± 0.07 | 7.98 ± 1.50 | 0.20 ± 0.12 | 0.14 ± 0.08 | 4.54 ± 0.44 | 0.41 ± 0.08 | 0.388 ± 0.11 |

Specifically, the performance was tested on Erdos-Renyi graphs (with varying edge probabilities). Figure 6 shows the MAE of reconstructing $\mathbf{A}$ for different graphs. The number of features for all graphs is set to $D = 10$, and their values are sampled from the standard normal distribution. Each subfigure represents different initialization strategies for $\hat{\mathbf{X}}$ and $\hat{\mathbf{A}}$. For each graph with a specific number of nodes, the MAE is averaged over 20 runs of the attack. As the number of nodes in a graph increases, the quality of reconstruction decreases. More importantly, it is evident that the vulnerability to the attack increases with decreasing sparsity. Fully connected graphs (ER graphs with edge probability 1.0) are the least vulnerable to the attack.

Table 19: Reconstruction performance (GSM-0, GSM-1, GSM-2, and Full) on Tox21 results using GCN.

| | GSM-0 | GSM-1 | GSM-2 | FULL |
|---|---|---|---|---|
| **GLG (Ours)** | $80.86^{+1.5}_{-1.2}$ | $96.51^{+1.1}_{-3.3}$ | $97.02^{+0.8}_{-1.1}$ | $98.71^{+1.0}_{-2.8}$ |
| **GRAIN** | $86.9^{+4.2}_{-5.7}$ | $83.9^{+5.2}_{-6.9}$ | $82.6^{+5.7}_{-7.4}$ | $68.0 \pm 1.7$ |
| **DLG** | $31.8^{+4.5}_{-4.3}$ | $20.3^{+5.5}_{-4.8}$ | $22.8^{+6.6}_{-5.6}$ | $1.0 \pm 0.2$ |
| **DLG + A** | $54.7^{+3.9}_{-4.2}$ | $60.1^{+4.6}_{-5.2}$ | $76.7^{+3.6}_{-4.8}$ | $1.0 \pm 0.2$ |
| **TabLeak** | $25.1^{+5.1}_{-4.3}$ | $12.4^{+5.5}_{-4.3}$ | $10.8^{+5.6}_{-3.9}$ | $1.0 \pm 0.2$ |
| **TabLeak + A** | $55.6^{+3.9}_{-3.9}$ | $57.7^{+4.1}_{-4.6}$ | $73.8^{+2.8}_{-3.5}$ | $1.0 \pm 0.2$ |

Table 20: Reconstruction of $\mathbf{X}$ (RNMSE $\downarrow$) and $\mathbf{A}$ (AUC $\uparrow$ and AP $\uparrow$) with Graph Attacker-GN on Tox21 dataset using GCN.

| Framework | $\mathbf{X} \downarrow$ | AUC $\uparrow$ | AP $\uparrow$ |
|---|---|---|---|
| **GLG (Ours)** | 0.76±0.13 | 0.93±0.10 | 0.94±0.04 |
| **GRAIN** | 1.35±0.03 | 0.91±0.10 | 0.65±0.11 |

## H   Comparison with Other Methods

In this section, we compare our proposed gradient inversion attack, GLG, against several existing methods, including the only other graph-specific gradient inversion approach, GRAIN (Drencheva et al., 2025), as well as two widely used baselines originally designed for image or tabular data, DLG and TabLeak. We evaluate our approach on the Tox21 dataset, a well-known chemical dataset from the MoleculeNet benchmark (Wu et al., 2018). The task is to reconstruct both the node features $\mathbf{X}$ and the adjacency matrix $\mathbf{A}$ from gradients alone.

Our evaluation utilizes the GSM-$k$ metric as introduced in Drencheva et al. (2025), where for each $k$ we aggregate the $k$-hop neighborhood for each node. This is done by randomly initializing a GCN model with $\geq k$ layers. We also include our evaluation metrics on the Tox21 dataset to provide a more detailed comparison.

### H.1   Detailed Analysis

We evaluate our method under the Graph attacker-GN threat model, which attempts to reconstruct both the node feature matrix $\mathbf{X}$ and the adjacency matrix $\mathbf{A}$ given the gradients. Table 19 shows that while reconstruction of $\mathbf{X}$ remains challenging, as indicated by a relatively lower GSM-0 score, reconstruction of the graph structure (reflected in the GSM-1, GSM-2 and FULL scores) is significantly improved compared to other methods. Table 20 presents the RNMSE, AUC, and AP results for a more detailed comparison between GLG and GRAIN.

**Observations.**

- **Node Feature Reconstruction (GSM-0):** The GSM-0 score measures the fidelity of the node feature reconstruction. Our experiments indicate that reconstructing $\mathbf{X}$ from gradients is challenging due to the lack of direct node-level supervision during backpropagation through graph convolutional layers.
- **Subgraph and Full Graph Reconstruction (GSM-1, GSM-2, FULL):** In contrast, the higher GSM-$k$ scores for $k \geq 1$ demonstrate that the graph structure is reconstructed with high accuracy. This suggests that even if individual node features are not perfectly reconstructed, the aggregation of multi-hop neighborhood information helps approximate the overall topology of the graph.
- **Comparison with GRAIN:** Although GRAIN is tailored for graph gradient inversion attacks, our method achieves higher scores in capturing global structure as indicated by improved GSM-1, GSM-2, and FULL metrics. Table 20 shows a similar trend: GLG more effectively reconstructs the graph structure, as indicated by higher AUC and AP, while also slightly improving nodal feature reconstruction, as reflected by the lower RNMSE.
- **Baseline Methods (DLG and TabLeak):** These methods, originally developed for images and tabular data, underperform in the graph domain. Even when augmented with the adjacency matrix information ("$+\mathbf{A}$"), their performance across all GSM metrics remains inferior, highlighting the unique challenges of gradient inversion attacks on graph data.

**Interpretation.**   The lower GSM-0 score suggests that reconstructing node features $\mathbf{X}$ purely from gradient information is non-trivial. However, the significant improvements observed in GSM-1, GSM-2, and FULL indicate that the model is highly effective in capturing the graph structure. This is critical, as many practical applications depend on the accurate reconstruction of the graph topology rather than on precise node-level details.

**Conclusion.** Overall, our experimental results demonstrate that our method outperforms existing gradient inversion attacks on the Tox21 dataset, particularly in reconstructing the global graph structure. Although node feature reconstruction remains an open challenge, leveraging multi-hop neighborhood information through the GSM-$k$ metric allows for robust approximation of the overall graph, thereby opening new avenues for research in graph-based gradient inversion attacks.

