# OpenReview forum: "Gradient Inversion Attack on Graph Neural Networks"
_TMLR — Accepted by TMLR_

### Review · Reviewer_mTSV · 2025-03-16

**Summary Of Contributions:**

This paper studies the gradient inversion attacks in federated graph learning to recover sensitive attributes such as node features as well as graph structures, depending on the specific classification tasks. This is the first work contributing to understand the privacy leakage in the vanilla graph federated learning. Empirical results show that the proposed approach indeed can leak significant information in different settings.

**Audience:**

Yes

**Broader Impact Concerns:**

Broader impact is not considered and the authors should discuss the implications of the proposed attacks to the society.

**Claims And Evidence:**

No

**Requested Changes:**

1. Clearly define the research challenges. And in response to this, the authors should show the results of the existing baselines in the Euclidean domain and show how (well/poorly) these methods would perform when applied with minimal or somewhat trivial modifications. I would expect the Eq. (13) would be a key deciding factor and without this term, the attack performance may be low on graph data. However, using such a term also makes the distinction with Zhang et al. (2022) unclear and the authors need to clearly position themselves well in the literature.
2. For the threat model, the authors should consider levels of threat models to understand the effectiveness of the attacks. It is fine to start from the complete white-box settings where the attacker knows the all possible information about the target victim. Then, start to relax some of the assumptions such as only having access to partial gradient information, or the adversarial clients are not always selected during the model training process. Instead of exhaustively listing all possible changes, the general guideline here is to consider what assumptions are likely too strong in practical applications and then gradually relax them to achieve a holistic view of the attack performance.

**Strengths And Weaknesses:**

Strengths:
1. Understanding the privacy leakage in graph federated learning is important.
2. The proposed approach achieved some reasonable effectiveness in inducing privacy leakage.

 Weaknesses:
1. The research challenge is not clearly defined. There are many gradient inversion attacks in other euclidean domains such as images. Therefore, a very natural question is, whether those methods can be effectively applied with minimal modifications. If the modifications are not straightforward and requires significant research effort, this can be claimed as a new contribution. This part is basically lacking from the paper.
2. The threat model is never mentioned in the paper. Such threat modeling is a standard practice in cybersecurity related problems and the authors should clearly define what is the attacker capability as well as the attacker knowledge.

---

> ### Author Response · Authors · 2025-04-05
> **Response to reviewer**
>
> We thank the reviewer for valuable feedback. We appreciate the reviewer’s insightful comments and have incorporated these clarifications into the revised manuscript.
>
> **Response to Weakness 1: Challenge and Contribution**
>
> Adapting gradient inversion attacks to graph data requires jointly reconstructing nodal features and graph structure. Hence, existing methods that focus on Euclidean data, i.e., DLG and its enhanced methods, are either infeasible or suffer substantial performance degradation when applied to graph data. In our revised experiment, we include DLG as a baseline and demonstrate that omitting our graph-specific regularization leads to poor reconstruction performance, see Tables 3,7,10 in the revised manuscript. Specifically, in GCN, DLG consistently struggles to reconstruct the graph structure (adjacency matrix A), which is essential for graph data.
>
> The GraphMI proposed by Zhang et al. (2022) is a model-inversion attack tailored to GNNs, which aims to reconstruct the adjacency matrix (graph structure) from the GNN Model. GraphMI is not designed for federated graph learning and operates under different assumptions: it has direct access to the models, nodal features and nodal labels instead of gradients.
>
> **Response to Weakness 2: Threat model**
>
> The threat model **was** provided in Section 4.1 (formerly Section 3.1 in original submission), detailing information accessibility for all 3 attacker scenarios across 7 different settings. For Node Attacker 2, 2-G can additionally access the information of nodal features ($\mathbf{X}$) in the subgraph and tries to recover the underlying the graph structure ($\mathbf{A}$); 2-N can additionally access the information of graph structure ($\mathbf{A}$) and tries to recover the nodal features ($\mathbf{X}$); 2-GN only can access the gradients of nodes and tries to recover both graph structure ($\mathbf{A}$) and nodal features ($\mathbf{X}$). Similarly, for graph attack, graph attacker-G has access to the gradients along with the nodal features ($\mathbf{X}$) and tries to recover graph structure ($\mathbf{A}$); graph attacker-N has access to the gradients along with the graph structure ($\mathbf{A}$) and tries to recover nodal features ($\mathbf{X}$); graph attacker-GN only has access to the gradients and tries to recover both nodal features ($\mathbf{X}$) and the graph structures ($\mathbf{A}$). For both node attacker 2 and graph attack, since attacker-G and N have additional information, the reconstruction performance is better than attacker-GN, as shown in Tables 5,6,7 for node attacker and Tables 8,9,10 for graph attacker.
>
> **Response to Broad Impact concerns:**
>
> We use only publicly available datasets and ensure no real-world privacy risks are incurred. Our work aims to raise awareness and promote the development of privacy-preserving frameworks tailored to graph-structured data in FGL. We highlight potential risks and hope our findings will encourage further research on effective defensive mechanisms, especially from graph perspectives.
>
> **Revision according to requested changes:**
>
> **Requested change 1**: We revised Section 1 to clarify the challenge and contributions. Related work is discussed in Section 2. In the revised manuscript, we also added DLG as the baseline in our experiments, see Tables 3,7,10. See also **Response to Weakness 1.**
>
> **Requested change 2**: We revised Section 4 (formerly Section 3 in the original submission) and added Table 2 to more clearly present the threat model. See also **Response to Weakness 2.**

---

> > ### Comment · Reviewer_mTSV · 2025-04-05
> >
> > Thanks for clarifying my concerns and also it is fault for the threat model part. Overall, my concerns are well addressed and I suggest the authors to include the current revisions.

---

> > > ### Author Response · Authors · 2025-04-10
> > >
> > > Thank you for your thoughtful feedback, it was helpful in improving our manuscript. We have revised the current version accordingly and will definitely incorporate your suggestions in further revisions as well.

---

### Review · Reviewer_CLWN · 2025-03-19

**Summary Of Contributions:**

The authors propose to do gradient inversion attacks to reconstruct graph data (features, structure or both) in a federated learning scenario. They provide theoretic results for GCN and GraphSage and demonstrate seemingly good empirical performance for node classification and graph classification.

**Audience:**

Yes

**Claims And Evidence:**

Yes

**Requested Changes:**

- discuss the mentioned baseline in great detail
- add baselines (as GRAIN does; straightforward application of related work)
- compare to GRAIN empirically

**Strengths And Weaknesses:**

Strengths:

- I like the clarity of the writing
- the propositions add theoretical insights
- the performance seems to be good
- an effect of the regularisation is examiend

Weaknesses:

- There exists work that studies attacking GNN federated learning [GRAIN: Exact Graph Reconstruction from Gradients (https://arxiv.org/abs/2503.01838) but on openreview since ICLR 2025 submission] So the claim that this is the first work is not true. Since the other work is open since September it is debatable whether it is concurrent
- There is no baseline in the empiric eval

---

> ### Author Response · Authors · 2025-04-05
> **Response to reviewer**
>
> We would like to thank the reviewer for the constructive review and address the comments point by point as follows.
>
> **Response to Weakness 1: Comparison with concurrent work GRAIN**
>
> We thank the reviewer for pointing out GRAIN, which is indeed the most relevant work to ours. We have revised the claim and no longer state that GLG is the first work to study the gradient inversion attack on graph data. Instead, we clarify that our contribution extends this line of research to the FGL setting, introducing a gradient inversion attack mechanism with fewer limitations, i.e., GLG, and providing theoretical analysis.
>
> We would like to clarify that our submission to TMLR (in Feb. 2025) predates the ArXiv version of GRAIN (in March 2025). In addition, our work was previously submitted to USENIX Security 24 (Summer Paper #438), and the current submission is a revised and improved version of the USENIX submission.
>
> We will next provide the comparison with GRAIN in detail and experimental results with DLG as the baseline.
>
>
>
> 1. Comparison with concurrent work, GRAIN
>
> GRAIN assumes that the degree of a node is used as a nodal feature for training GNNs and is known by the attacker. In the step-by-step reconstruction, the degrees are critical to filtering, DFS reconstruction, and span check. By leveraging the degree information, GRAIN can significantly reduce the search space. Without knowledge of degrees, ensuring the theoretical or experimental results becomes uncertain. In their ablation study, removing the node's in-degree from the feature set causes substantial degradation. Our method, GLG, does not require this assumption and can **jointly** reconstruct both nodal features and graph structure. By employing the regularization, we offer broader applicability and highly accurate reconstruction of adjacency and nodal features.
>
>
> **Response to Weakness 2: Lack of baseline**
>
>
> To provide an empirical comparison with GRAIN, we apply GLG to the tox21 dataset. We conduct the test on GCN as both GRAIN and GLG are tailored to it. After the reconstruction, we compute the GSM matrix proposed by GRAIN to evaluate GLG performance. The results are shown in the table below. Note that the results for GRAIN and other baselines are directly adopted from their results.
>
> Experimental results comparing the performance of various methods using the GCN model on the tox21 dataset (adjacency reconstruction) NOTE: need to align the accuracy results with GSM
>
>
> | Method        | GSM-0 | GSM-1 | GSM-2 |
> |---------------|-------|-------|-------|
> | GLG           | 80.86 | 96.51 | 97.02 |
> | GRAIN         | 86.9  | 83.9  | 82.6  |
> | DLG           | 31.8  | 20.3  | 22.8  |
> | DLG +A        | 54.7  | 60.1  | 76.7  |
> | TabLeak       | 25.1  | 12.4  | 10.8  |
> | TabLeak +A    | 55.6  | 57.7  | 73.8  |
>
>
> It can be observed that even without knowledge of node degree, GLG can still outperform all baselines and achieve similar or better performance than GRAIN. The aforementioned results and analysis with more details are included in Appendix F of our revised manuscript.
>
>
> In the revised manuscript, we also added DLG as the baseline in our experiments on the GitHub and Facebook datasets, see Tables 3,7,10. We plan to incorporate it as a baseline across all experiments in our final submission. In future versions, we also aim to provide further comparisons across various datasets (e.g., CiteSeer, Pokec, Github, and FacebookPagePage) and evaluate performance using different metrics (i.e., RNMSE, AUC, and GSM).
>
> **Revision according to requested changes:**
>
> **Requested change 1**: We added the discussion about baselines in Section 2 and provided a detailed comparison with GRAIN in Appendix F. See also **Response to Weakness 1.**
>
> **Requested change 2 and 3**: We added DLG and GRAIN as new baselines, see Appendix F and Tables 3,7,10,19 in the revised manuscript, and the **Response to Weakness 2**.

---

> > ### Comment · Reviewer_CLWN · 2025-04-16
> >
> > I thank the authors for their rebuttal and I like the changes and I believe they are necessary for acceptance.

---

> > > ### Author Response · Authors · 2025-04-19
> > >
> > > Thank you for the helpful feedback. Following our initial revision, we have now incorporated DLG as a baseline across all experiments, including Node Attacker 1, Node Attacker 2, and Graph Attacker settings. Accordingly, we have updated Tables 5, 6, 8, and 9 in the revised manuscript to reflect these additions.

---

### Review · Reviewer_f6ne · 2025-03-21

**Summary Of Contributions:**

This paper investigates the vulnerability of Graph Neural Networks (GNNs) against gradient inversion attacks within the context of federated learning, where the goal is to train over large graph datasets while preserving data privacy. The motivation stems from the increasing use of GNNs in various applications and the potential risk of private data leakage through gradient exchanges, which has not been extensively studied for graph data. The authors propose a novel attack named Graph Leakage from Gradients and analyze its effectiveness on two widely-used GNN frameworks: GCN and GraphSAGE. The method involves both theoretical analysis and empirical validation to demonstrate that parts of the graph data can indeed be leaked from the gradients. The experiments are conducted over real-world social network datasets as well as synthetic datasets, using various evaluation metrics such as Root Normalised Mean Squared Error for nodal features and accuracy, AUC, and AP for graph structure recovery. The results show that attackers can recover significant information about the graph structure and nodal features.

**Audience:**

Yes

**Claims And Evidence:**

Yes

**Requested Changes:**

Overall, I think the paper needs further clarifications on comparisons with the liteature, novelty in model, and experimental designs (see Weaknesses for details).

**Strengths And Weaknesses:**

Strengths
1.	The paper studies gradient inversion attacks for graph federated learning, which is valuable for the privacy of graph machine learning.
2.	The paper is generally clearly written and easy to follow.
3.	The authors have provided theoretical support for their proposed method.

Weaknesses
1.	The authors state in the third paragraph of the introduction, "To our best knowledge, this is the first work that studies gradient-based attacks for graph data." However, they should clarify that while this may be the first study focusing on the federated learning scenario, numerous papers have already addressed gradient-based attacks in other contexts, such as PRBCD [1].
2.	The authors claim that few studies have investigated the issue of information leakage for graph data in the federated domain. However, based on my understanding, there is already a substantial body of existing research on privacy protection in the field of federated graph learning, e.g., see a comprehensive survey [2]. The authors should clearly compare with the literature to truly demonstrate the novelty of their contribution.
3.	For the technical part, my major concern is that the proposed method (Section 3) essentially adapts existing general-purpose gradient inversion attack methods into GNN scenarios with limited novelty. It would make the paper stronger if the authors could provide tailored model designs for federated graph learning. For example, the authors mention that graphs have structure and features, making them more difficult to extract. However, the proposed method seems not to delve deep into further analyzing and dealing with this challenge, but rather extracting features and structure separately.
4.	In experiments, the authors only verify the validity of the proposed method, but do not compare it with any baseline, which makes it hard to truly assess the effectiveness of the proposed method. Besides, the adopted datasets are not well-adopted benchmarks in graph machine learning (e.g., the Open Graph Benchmark). The authors may want to provide clarifications for the experimental designs.

[1] Robustness of graph neural networks at scale, NeurIPS’21
[2] A review of privacy-preserving research on federated graph neural networks, NeuroComputing’21

---

> ### Author Response · Authors · 2025-04-05
> **Response to reviewer**
>
> We would like to thank the reviewer for the constructive review and address the comments point by point as follows.
>
> **Response to Weakness 1:**
>
>
> We acknowledge that gradient-based attacks have been explored. The attack studied in this work specifically refers to gradient inversion, which aims to reconstruct data from gradients, rather than general gradient-based attacks. Most existing studies on gradient inversion attacks primarily focus on Euclidean data. In previous work, e.g.,  PRBCD [1], the gradient-based attack objective in FGL is to degrade model accuracy instead of reconstructing the data (features and topology). Graph-specific gradient inversion attacks, i.e.,  reconstructing both node features and graph structure from gradients, remain underexplored. Although the FGL framework is vulnerable to gradient-based attacks[2], the privacy-preserving methods designed to defend against gradient-inversion attacks in FGL remain unexplored. We have clarified this distinction and made modifications in the revised manuscript.
>
> **Response to Weakness 2:**
>
>
> While there has indeed been a substantial body of research on privacy protection in the field of federated graph learning, most works either do not attack based on the gradient or do not try to reconstruct the graph data, i.e., graph structure and nodal features. We have clarified this distinction and made modifications in the revised manuscript, emphasizing that our attack targets gradient inversion for both graph structure and nodal features, see Section 2 and Table 1. The gradient inversion attack on federated graph learning remains underexplored compared to other contexts.
>
> **Response to Weakness 3:**
>
>
> We have added a tailored objective function especially for graph data in Section 4 which utilises feature smoothness that relies on the adjacency and the node feature matrix being recovered. The threat model is provided in Section 4.1 (formerly Section 3.1), with information accessibility for all 3 attacking scenarios across 7 different settings. Among which, Node attacker 2-GN and Graph attacker-GN **jointly** recover both features and structure. We provide a detailed introduction to our attack mechanism in Section 4.2.
>
>
> Existing methods that focus on Euclidean data, i.e., DLG, cannot directly handle graph-structured data well due to interdependencies between nodal features and graph topology. Specifically, DLG performs poorly in reconstructing the graph structure (adjacency A), which is essential for graph data, see Tables 3,7,10. The graph structure encodes the relational topology of a graph and defines how nodes interact and influence each other. It not only influences the feature propagation and task performance, but also contains significant relational information, e.g., friendships in social networks and bonds in molecules.  We adapt the DLG as the baseline for experiments.
>
> **Response to Weakness 4:**
>
>
> We have added DLG on graph data as the baseline model for some of the experiment settings. Additionally, we have also added comparisons with other methods including GRAIN in Appendix F. The real datasets, GitHub and Facebook Page-Page, are widely used as social network benchmarks. We evaluate our method on them with an additional synthetic dataset across all seven settings defined in Section 4, with results presented in Sections 6.1–6.3. We focus more on the social network scenario, whereas Open Graph Benchmark (OGB) is not ideal for social network analysis. Additionally, the results for the molecular benchmark TUDatasets are provided in Appendix E. We will surely include more rigorous comparisons and also add the results on OGB for the future version.
>
> **Revision according to requested changes:**
>
>
> We revised Section 1 to clarify the challenge and contributions. Related work is discussed in Section 2. In the revised manuscript, we also added DLG as the baseline in our experiments, see Tables 3,7,10. Comparisons with other existing methods, including GRAIN added in Appendix F.
>
> [1] Robustness of graph neural networks at scale, NeurIPS’21 \
> [2] A review of privacy-preserving research on federated graph neural networks, NeuroComputing’21
>
> [3] Chaoyang He, Keshav Balasubramanian, Emir Ceyani, Yu Rong, Peilin Zhao, Junzhou Huang, Murali Annavaram, and Salman Avestimehr. Fedgraphnn: A federated learning system and benchmark for graph neural networks. CoRR, abs/2104.07145, 2021.
>
> [4] Simone Scardapane, Indro Spinelli, and Paolo Di Lorenzo. Distributed training of graph convolutional networks. IEEE Transactions on Signal and Information Processing over Networks, 7:87–100, 2020.
>
> [5] Huanding Zhang, Tao Shen, Fei Wu, Mingyang Yin, Hongxia Yang, and Chao Wu. Federated graph learning–a position paper. arXiv preprint arXiv:2105.11099, 2021.

---

### Decision · Action_Editor_VQD5 · 2025-05-19

**Recommendation:** Accept with minor revision

**Comment:**

This paper investigates the vulnerability of Graph Neural Networks (GNNs) to gradient inversion attacks in federated learning, with the goal of training on large graph datasets while preserving data privacy. The motivation is strong, and the authors propose a novel attack method, Graph Leakage from Gradients, to analyze GCN and GraphSAGE. The paper provides both empirical evaluations and theoretical analyses, and the writing is clear and well-organized. There are a few suggestions for improvement: it would be beneficial to include baseline results, clarify some claims and statements, and better illustrate the challenges and rationale behind the method design. Overall, this is a good work that could be accepted after revision.

**Audience:**

The paper will attract the audience in the field of graph machine learning.

**Claims And Evidence:**

The evidence is accurate and convincing.